# Integration of time-series meta-omics data reveals how microbial ecosystems respond to disturbance

Malte Herold [1,2], Susana Martínez Arbas [1], Shaman Narayanasamy[1,3], Abdul R. Sheik[1], Luise A. K. Kleine-Borgmann[1], Laura A. Lebrun[1], Benoît J. Kunath [1], Hugo Roume[1,4], Irina Bessarab[5], Rohan B. H. Williams [5], John D. Gillece[6], James M. Schupp[6], Paul S. Keim [6], Christian Jäger[1], Michael R. Hoopmann[7], Robert L. Moritz [7], Yuzhen Ye[8], Sujun Li [8], Haixu Tang[8], Anna Heintz-Buschart [1,9,10], Patrick May [1], Emilie E. L. Muller [1,11], Cedric C. Laczny [1] & Paul Wilmes [1,12 ✉]

The development of reliable, mixed-culture biotechnological processes hinges on understanding how microbial ecosystems respond to disturbances. Here we reveal extensive phenotypic plasticity and niche complementarity in oleaginous microbial populations from a biological wastewater treatment plant. We perform meta-omics analyses (metagenomics, metatranscriptomics, metaproteomics and metabolomics) on in situ samples over 14 months at weekly intervals. Based on 1,364 de novo metagenome-assembled genomes, we uncover four distinct fundamental niche types. Throughout the time-series, we observe a major, transient shift in community structure, coinciding with substrate availability changes. Functional omics data reveals extensive variation in gene expression and substrate usage amongst community members. Ex situ bioreactor experiments confirm that responses occur within five hours of a pulse disturbance, demonstrating rapid adaptation by specific populations. Our results show that community resistance and resilience are a function of phenotypic plasticity and niche complementarity, and set the foundation for future ecological engineering efforts.

[1] Luxembourg Centre for Systems Biomedicine, University of Luxembourg, 7 Avenue des Hauts-Fourneaux, 4362 Esch-sur-Alzette, Luxembourg, Luxembourg. [2] Epidemiology and Microbial Genomics, Laboratoire National de Santé, 1 rue Louis Rech, 3555 Dudelange, Luxembourg. [3] Megeno S.A., 6A Avenue des Hauts-Fourneaux, 4362 Esch-sur-Alzette, Luxembourg. [4] MetaGenoPolis, Institut National de Recherche pour l'Agriculture, l'Alimentation et l'Environnement, Université Paris-Saclay, Domaine de Vilvert, Bâtiment 325, 78350 Jouy-en-Josas, France. [5] Singapore Centre for Environmental Life Sciences Engineering, 60 Nanyang Dr, Singapore 637551, Singapore. [6] The Translational Genomics Research Institute, 3051 West Shamrell Boulevard, Flagstaff, AZ 86001, USA. [7] Institute for Systems Biology, 401 Terry Avenue North, Seattle, WA 98109, USA. [8] School of Informatics, Computing and Engineering, Indiana University, 700 N. Woodlawn Avenue, Bloomington, IN 47405, USA. [9] German Centre for Integrative Biodiversity Research (iDiv) Halle-Jena-Leipzig, Puschstr. 4, 04103 Leipzig, Germany. [10] Helmholtz Centre for Environmental Research GmbH – UFZ, Theodor-Lieser-Str. 4, 06120 Halle, Germany. [11] Equipe Adaptations et Interactions Microbiennes, UMR 7156 UNISTRA-CNRS, Université de Strasbourg, Strasbourg, France. [12] Department of Life Sciences and Medicine, Faculty of Science, Technology and Medicine, University of Luxembourg, 7 Avenue des Hauts-Fourneaux, L-4362 Esch-sur-Alzette, Luxembourg. ✉email: paul.wilmes@uni.lu

Mixed-culture biotechnological processes are essential for humankind to achieve its sustainable development goals[1,2]. However, in order to engineer reliable processes, fundamental insights into microbial niche ecology are necessary. Biological wastewater treatment plants (BWWTPs) represent a ubiquitous biotechnological application and occupy a central position in sustainable resource management plans[3,4]. Oleaginous bacterial populations are commonly found as the main constituents of floating sludge in BWWTPs and include divergent taxa such as Candidatus Microthrix parvicella or Acinetobacter spp.[5]. Storage lipids, such as triacylglycerols (TAGs), wax esters (WEs), and polyhydroxyalkanoates (PHA), derived from the lipid-rich biomass can directly be transesterified to fatty acid alkyl esters (biodiesel)[5], whereby PHA also represents a suitable precursor for bioplastics[6]. In general terms, substrate storage provides microbial populations with a competitive advantage under rapidly fluctuating and oftentimes sparse substrate conditions[7,8]. Even though BWWTP operation is a controlled process, factors such as aeration cycles, seasonal changes in temperature, and composition of inflow wastewater fluctuate. These factors have a profound impact on population dynamics[9] as well as linked process efficiency[10]. For example, periods of inefficient operation have been linked to competition between polyphosphate and glycogen accumulating organisms[11]. However, for wastewater-borne lipid-accumulating populations, which have compelling potential to be used in circular economic models[3], community shifts have been observed[12–14] with yet unclear links to niche ecology in situ.

Integrated meta-omics approaches hold the potential to resolve the fundamental niches and realized niches of microbial populations in situ[15]. The former represents the exhaustive inventory of resource ranges and conditions sustaining viability in the absence of environmental stress, competition, or predation, while the latter represents the part of a fundamental niche that is actually utilized by a population in the presence of other species and in a particular environment. The reconstruction of the fundamental niches is possible by linking functional potential to metagenome-assembled genomes (MAGs)[16] obtained through metagenomic (MG) sequencing. Functional omics data, such as metatranscriptomics (MT) or metaproteomics (MP), allow the resolution of realized niches[16]. Meta-omics approaches have previously been used for comparative functional screening in different environments and to characterize microbial activity, e.g., by using MT/MG ratios[17,18]. In human gut-borne microbial communities, niche partitioning has been inferred based on transcriptional profiles[19]. Furthermore, the coupling of MT and MP to meta-metabolomic (MM) data allows the differentiation between niches of genetically closely related populations[20]. Resolving the functions of coexisting microbial populations is of particular interest in the context of the extensive functional redundancy within microbial ecosystems[21,22]. Based on their emergent properties[23], microbial communities are characterized by composite metabolic capabilities and increased robustness compared to individual strains[24,25]. Steering these complex systems towards a desired endpoint, e.g., increased lipid accumulation, requires in-depth understanding of niche space and stability.

Here, we study whether community resistance and resilience are a function of phenotypic plasticity and niche complementarity. We develop and apply a novel framework for the in situ characterization of fundamental and realized niches of individual populations providing an in-depth understanding of ecological processes within a microbial community. We delineate ecological niches by integrating longitudinal meta-omics data (MG, MT, MP, and MM) and study complementarity of the realized niches. The addition of functional omics data (MT, MP, and MM) enables the resolution of metabolic plasticity and we thereby reveal how microbial ecosystems respond to disturbance. Using ex situ experiments to simulate pulse disturbances, we assess the response of individual oleaginous populations to oleic acid addition under shifting dissolved oxygen concentrations. Our dataset and methods represent important resources for the emerging field of integrating meta-omics data to study mixed microbial communities. Our results contribute to applications beyond wastewater treatment such as informed ecological engineering or research on host-associated microbiota.

## Results

**A time-resolved meta-omics dataset**. To characterize the niche space of lipid-accumulating populations as well as resistance and resilience of the microbial community, we sampled a municipal BWWTP weekly over a 14-months period (from 2011-03-21 to 2012-05-03). Additionally, two preliminary time-points outside of the time-series were included[13,26]. Samples were split into intracellular and extracellular fractions, followed by concomitant biomolecular extractions[27] and high-throughput measurements (Fig. 1). MG, MT, and MP data were obtained on the intracellular fractions and MM data was generated on both the intracellular and extracellular fractions.

After quality filtering, the per-sample averages of MG and MT reads were $5.3 \times 10^7$ ($\pm 7.7 \times 10^6$ s.d.) and $3.3 \times 10^7$ reads ($\pm 1.2 \times 10^7$ s.d.), respectively (Supplementary Data 1). We performed sample-specific genome assemblies (average of $4.1 \times 10^5$ contigs per sample) followed by binning[28] yielding a total of 1364 MAGs passing our quality filtering criteria (see "Methods" section). To track the abundance, gene expression, and activity of individual microbial populations over time, we dereplicated[29] the MAGs across samples to generate 220 representative MAGs (rMAGs). From these, we further selected those with the highest completeness resulting in 78 rMAGs (76.2% mean completeness, 2.2% mean contamination) (Supplementary Data 2). These genomes represent the major populations across the time-series, with an average mapping percentage of 26% ± 3% (s.d.) and 27% ± 3% (s.d.) of total MG reads and total MT reads per time-point, respectively, and are corroborated by a previous study based on 16S rRNA amplicon sequencing[13]. For the MP measurements, we obtained a per-sample average of $1.5 \times 10^5 \pm 8.2 \times 10^3$ (s.d.) MS2 spectra and a total of $7.6 \times 10^6$ MS2 spectra. Of $7.8 \times 10^5$ identified peptides, $3.3 \times 10^5$ (43%) could be matched to $2.1 \times 10^5$ predicted coding sequences of the 78 rMAGS. Per time-point, on average $1.5 \times 10^4 \pm 4.5 \times 10^3$ (s.d.) spectral matches, i.e., on average 94% of all rMAG-associated matches could be assigned to genes with predicted functions, i.e., assigned KEGG ortholog groups (KOs). To study the community-wide resource space and metabolite turnover, we measured metabolite levels by an untargeted approach using gas chromatography (GC) coupled with mass spectrometry (MS) (Supplementary Data 3). In total, 89% (58 of 65) of the identified metabolites could be linked to enzymes encoded by the rMAGs. We estimated resource uptake by calculating intracellular vs. extracellular metabolite ratios for 42 metabolites detected in both fractions (Supplementary Data 3). Additionally, six abiotic parameters were measured during sampling, as well as 34 parameters recorded continuously as part of the BWWTP online monitoring (Supplementary Data 4).

We also generated MG and MT data for the ex situ experiments. These simulated the fluctuating conditions within the BWWTP, namely the short-term response to pulse disturbances of oleic acid influx under shifting dissolved oxygen conditions. We sequenced DNA and RNA fractions obtained at 0, 5, and 8 h after addition of oleic acid, yielding on average $1.02 \times 10^8$ MG and $9.33 \times 10^7$ MT reads per sample. The increased sequencing depth compared to the in situ time-series was

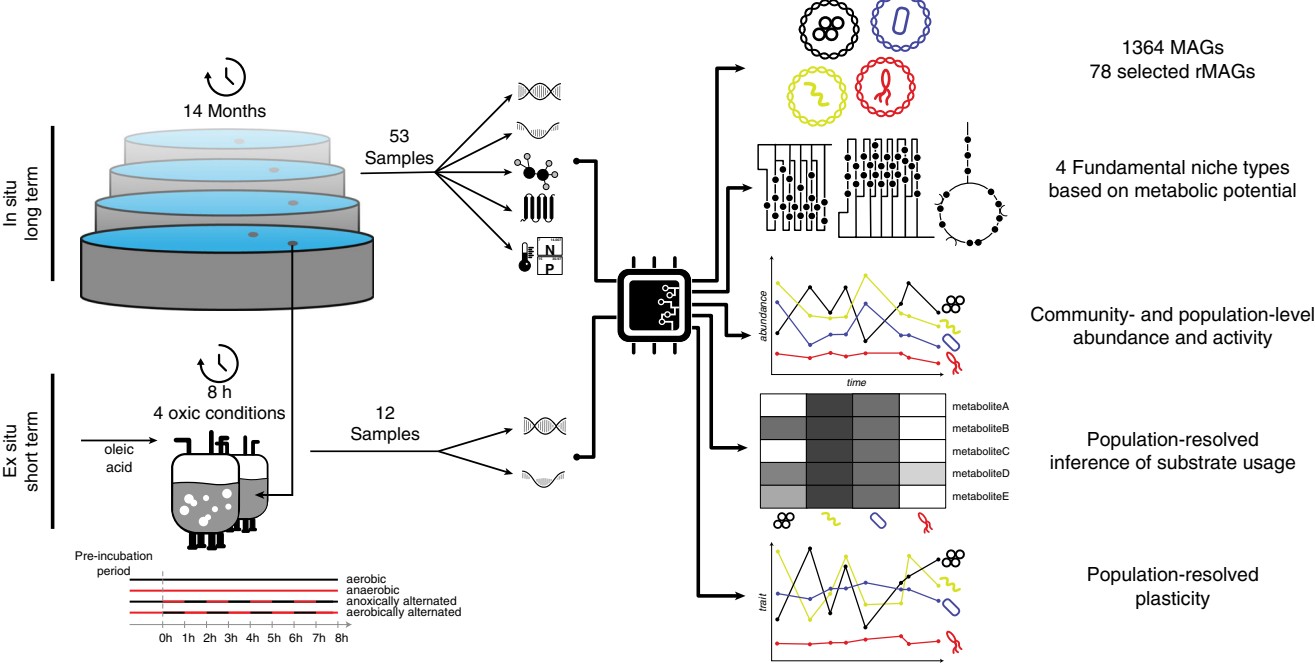

**Fig. 1 Overview of the study design.** Samples are derived from in situ sampling of an anoxic tank of a municipal biological wastewater treatment plant. Metagenomic (MG), metatranscriptomic (MT), metaproteomic (MP), and meta-metabolomic (MM) data is generated. Physicochemical data is also collected. Additionally, MG and MT data is generated for ex situ experiments using biomass from the same system and fed with oleic acid under different oxic conditions to evaluate short-term responses to pulse disturbance. The time-series meta-omics data is integrated to define metagenome-assembled genomes (MAGs) over all time points. Representative MAGs (rMAGs) across time are selected for further analysis. The rMAGs' functional potential is used to infer the fundamental niches. Abundance and activity data are derived from the functional omics and substrate usage is inferred per population. The variability of gene expression is used to assess the phenotypic plasticity of the individual populations.

important to obtain a fine-grained view on short-term responses to oleic acid. Mapping of the sequencing reads to the selected set of rMAGs revealed mapping percentages comparable to the in situ time-series (mean: 21% ± s.d.: 1% for both MG reads and MT reads).

Overall, our meta-omics dataset comprehensively describes mixed microbial communities underlying lipid-accumulation processes in BWWTPs, and in particular their functional potential, composition, activity, as well as substrate availability and assimilation.

**Distinct niche types**. To resolve the fundamental niches of the pertinent bacterial populations through their functional genomic potential, we assigned KOs to the rMAGs' predicted coding sequences. We hypothesized that individual populations would form clusters based on the similarity/dissimilarity of their functional potential. We found four distinct clusters of rMAGs by projecting pairwise Jaccard distances of KO presence (Fig. 2a and Supplementary Fig. 1). These functional clusters (FunCs) represent differences of known, overall metabolic capabilities of the rMAGs and reflect their fundamental niches. FunC-1 consisted of Actinobacteria, and FunC-2 was primarily comprised of members of the Bacteroidetes phylum, mainly of the Sphingobacteriia class (Fig. 2a). FunC-3 contained Betaproteobacteria and Gammaproteobacteria whereas FunC-4 appeared more diverse, containing Spirochaetia as a subcluster, Deltaproteobacteria, and taxonomically unclassified rMAGs. We found mash-based genomic distance[30] to be strongly linked to FunC assignment (PROCRUSTES sum of squares: 0.399, correlation 0.775, PROTEST p-value 0.001, Supplementary Fig. 2a), highlighting that phylogeny is a strong determinant for FunC assignment. However, some

distantly linked subgroups were defined by their shared functional complement, i.e., assigned to a different FunC than their neighbors in a corresponding phylogenetic tree (Supplementary Fig. 2b). This shows that KO profile similarity-based analyses provide important information in addition to phylogeny-based approaches[31].

A total of 1857 KOs was shared between all FunCs and we found that FunCs 1, 3, and 4 contained comparable numbers of nonredundant KOs with 4276, 4177, and 4129 KOs, respectively (Fig. 2b). FunC-2 exhibited a reduced number of KOs (3550), however it also represented the least taxonomically diverse FunC as it almost exclusively consisted of *Haliscomenobacter* spp. and *Chitinophaga* spp. (Supplementary Data 2). We tested for the molecular functions that were significantly enriched in individual FunCs and found, among others, functions related to lipid metabolism for FunC-1, amino sugar, and nucleotide sugar metabolism for FunC-2, and biofilm and secretion systems for FunC-3 to be enriched (Fig. 2c and Supplementary Data 5; one-sided Fisher's exact test, adjusted p-values < 0.05).

While lipid-accumulating organisms hold great potential for the recovery of high-value molecules[5], interactions between these organisms as well as the community at large are understudied in situ. We found that diacylglycerol O-acyltransferase (DGAT/WS), which is involved in lipid storage[32], was encoded in 23 out of 24 rMAGs of FunC-1, pointing to the importance of TAG accumulation in this cluster. Most FunC-3 members also encoded DGAT/WS (14 of 19). Moreover, PHA synthase was enriched in this cluster (15 of 19). All rMAGs encoded lipases, functions involved in fatty acid synthesis, or beta-oxidation. However, several acyl-CoA and acyl-ACP dehydrogenases were overrepresented in FunC-1 and FunC-3. Additionally, acetyl-CoA acetyltransferases involved in the degradation and biosynthesis of

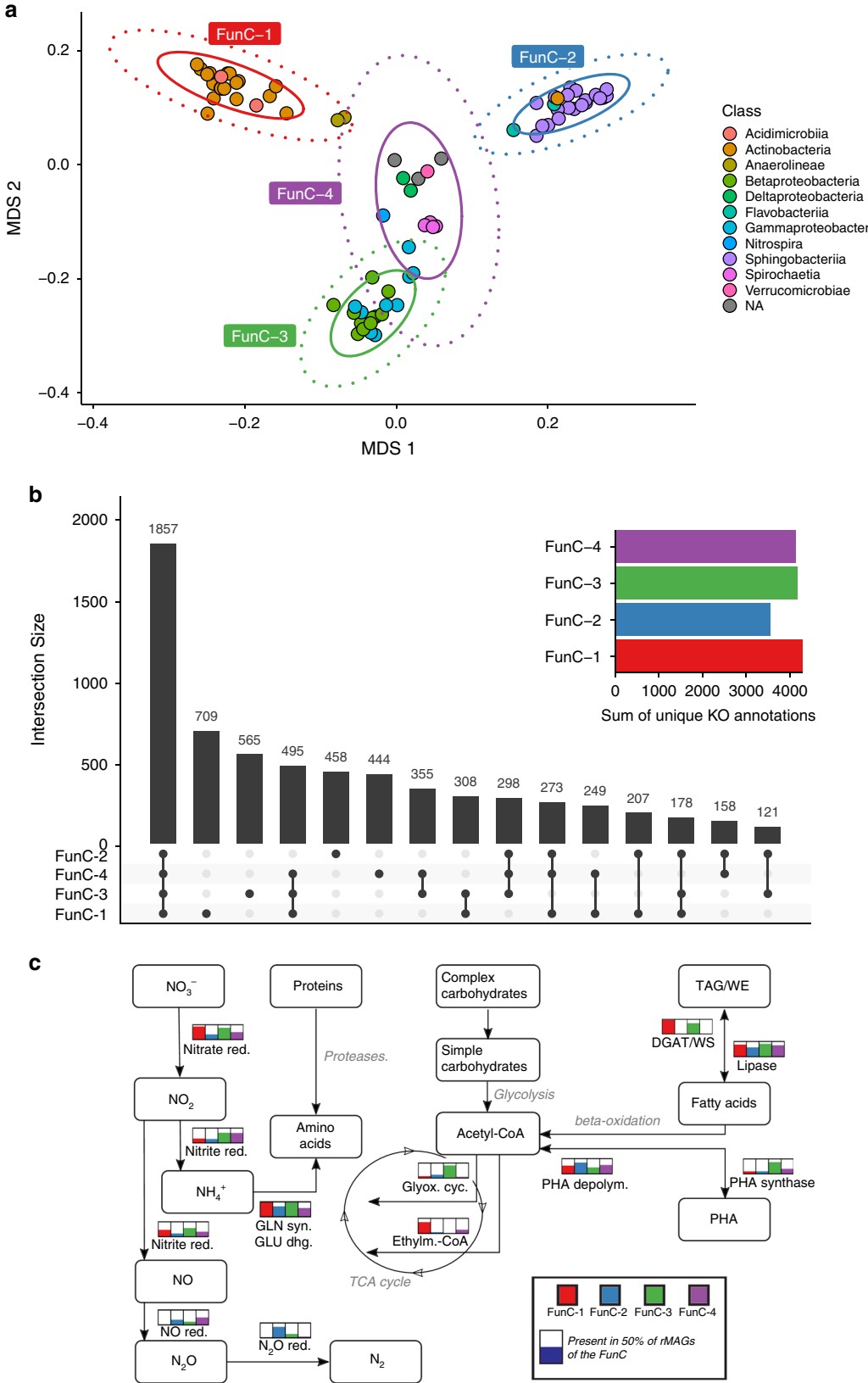

fatty acids were prevalent throughout all FunCs. The enrichment in FunC-1 and FunC-3 for genes involved in lipid accumulation are consistent with previous metabolic characterizations, with FunC-1 consisting mainly of Actinobacteria for which TAG accumulation has been described[33]. FunC-3 contains Betaproteobacteria and Gammaproteobacteria that have been

characterized as TAG, WE, and/or PHA accumulators, e.g., *Thauera* spp., *Albidiferax* spp., or *Acinetobacter* spp.[33,34]. Importantly, we observed a difference between these FunCs in the utilization of acetyl-CoA. Specifically, FunC-1 members showed an enrichment in functions related to the ethylmalonyl-CoA pathway (crotonyl-CoA reductase and enoyl ACP

**Fig. 2 Fundamental niche types. a** Multidimensional scaling (MDS) of Jaccard distances for the functional repertoire (presence of KEGG ortholog groups [KOs]) for each rMAG. Ellipses containing 95% (inner) or 99% (outer) of cluster-assigned data points are shown resulting in four distinct functional clusters (FunCs). Colors indicate the class-level taxonomy of the rMAGs. **b** Numbers of shared and unique KO assignments between the FunCs. Colored bars show the total number of nonredundant KO assignments within the individual FunCs. Overlaps between different sets of FunCs and their unique KOs are represented by the central black bars with the points below defining the members of the respective sets. **c** Presence of key functions within the four FunCs. Bars next to metabolic conversions show the proportion of rMAGs encoding characteristic enzymes for the respective reaction or pathway adjusted for mean rMAG completeness. Pathways ubiquitously present across rMAGs are shown in gray color. Source data are provided as a Source Data file. red. reductase, GLN syn. glutamine synthetase, GLU dhg. glutmatate dehydrogenase, glyox. cyc. glyoxylate cycle, ethylm.-CoA ethylmalonyl-CoA pathway, PHA depolym. PHA depolymerase.

reductase), while FunC-3 members encoded key enzymes involved in the glyoxylate cycle (malate synthase and isocitrate lyase).

We further determined specific functional enrichment for the four FunCs in relation to the breakdown of other macromolecules (including CAZymes and proteases), nitrogen cycling, stress response, and motility (Supplementary Data 5). The discriminating functions point towards interdependencies between the different FunCs, e.g., in terms of denitrification (Fig. 2c). We found that the separation into taxonomically consistent groups is accompanied by specific conserved functions, e.g., strong enrichment in FunC-1 for WhiB transcriptional regulators characteristic of the Actinobacteria[35]. Overall, we observed a widely distributed set of core functions in foaming sludge microbiomes and identified groups of populations characterized by distinct functional potential in lipid metabolism, amino sugar, and nucleotide sugar metabolism as well as biofilm and secretion systems.

**Community dynamics and stability.** To understand whether population dynamics can be related to substrate availability and other abiotic factors[36], we used MG depth-of-coverage to infer rMAG population abundance across the time-series. We computed distances between the rMAGs' abundance profiles (based on their pairwise correlations) and found that the dynamics of rMAGs can be partially explained by the FunC assignment (PERMANOVA $R^2 = 0.12$, Pr > F = 0.002; no significant difference in dispersion; Supplementary Fig. 3), thereby linking FunC membership to temporal abundance shifts. The most abundant taxa (Supplementary Data 2) included Candidatus Microthrix (26.0% relative abundance across the time-series; referred to as Microthrix in the remaining text), Acinetobacter (8.1%), Haliscomenobacter (8.0%), Intrasporangium (7.2%), Leptospira (6.3%), Albidiferax (5.7%), and Dechloromonas (2.4%) (Fig. 3a). Several of the recovered rMAGs belonged to filamentous taxa according to the MiDAS field guide database for organisms in activated sludge[37], such as the highly abundant Microthrix, and Haliscomenobacter, as well as the less abundant Anaerolinea (1.1 %) and Gordonia (0.2 %).

Variations during the operation of BWWTPs occur largely due to changes in the influent wastewater composition and climatic conditions[38]. We observed gradual changes in the community structure with the seasons (Fig. 3a). In October 2011 (month seven of the timeseries), the community composition began to shift, with a markedly altered composition in late November 2011. This shift was characterized by spikes in the relative abundance of Leptospira (peak at 2011-11-23) and Acinetobacter (peak at 2011-11-29) (Fig. 3a), and co-occurred with a pronounced shift in substrates (Fig. 4 and Supplementary Fig. 4). The substrates included mainly nonpolar metabolites, including long-chain fatty acids (LCFAs) and glycerides, as well as polar metabolites mannose, glucose, disaccharides, ethanolamine, and putrescine. We found that the intersample distances of MG-based abundances could partially be explained by a subset of the abiotic

factors (Fig. 3c). Summer samples were characterized by higher temperatures, phosphate levels and higher intracellular vs. extracellular oleic acid ratios. Higher extracellular mannose levels and a slight increase in conductivity marked the beginning of the autumn shift. During November, intracellular and extracellular levels for LCFAs increased, indicating a higher availability or turnover of LCFAs, but not necessarily an equivalent conversion to neutral storage lipids. In the subsequent winter time-points, substrate levels normalized and the community transitioned back to the predisturbance state.

The dominance of Microthrix was re-established within approximately ten generations, given estimates for in situ growth rates of 0.12–0.3 growth cycles per day[7,8]. The stability[39] of the individual rMAGs was heavily affected by the November shift (mean population stability: $1.43 \pm 0.69$ s.d.), compared to the stability when excluding the respective time-points (mean population stability: $2.39 \pm 1.28$ s.d.; 2011-11-02 to 2011-11-29; Supplementary Data 2). The observed population dynamics indicate that the community composition is resilient, i.e., recovers after pronounced changes in available substrates, and resistant to small-scale environment fluctuations over time.

While MG depth was used as a proxy for population abundance, MT depth enabled the analysis of transcriptional activity within the community and of individual populations (Fig. 3b). The comparison of intersample distances based on mean, relative MT depth showed similar patterns to MG-based results (Fig. 3c), albeit with a higher degree of variability indicated by increased inter-sample distances (Fig. 3d). A comparison of relative MP counts showed a more even distribution between populations with comparable overall trends (Supplementary Fig. 5). Samples collected in April 2011 and 2012 appeared to represent transition states between seasons. Additionally, a set of late winter and early spring samples in 2011 and 2012 showed higher similarities at the expression level than at the abundance level. Interestingly, the high abundance of individual genera, such as Microthrix or Chitinophaga was not necessarily reflected in their mean expression levels (Fig. 3b and Supplementary Fig. 5): populations assigned to Leptospira, Haliscomenobacter, Anaerolinea, and Acinetobacter showed higher mean expression overall. Spikes in relative MT depth as for Acinetobacter rMAGs (Fig. 3b; 2011-04-14, 2011-05-08, and 2012-04-25) point towards increased activity around these time-points, which however did not lead to major shifts in community structure. Notably, higher expression levels of Acinetobacter were succeeded by increased expression levels of Haliscomenobacter (2011-04-14 to 2011-05-20) or Anaerolinea (2011-05-08 to 2011-09-19). On average, MT-based stability values were less affected by the community shift than MG-based stability values (Supplementary Table 2). We also observed adaptation of metabolic pathway activity to environmental conditions (Fig. 5). Pentose to EMP pathway intermediates exhibited the highest correlation between MT and MP abundances, followed by Hydrogen metabolism and Fatty acid oxidation. Several pathways exhibited a characteristic drop during the November shift, e.g., hydrogen metabolism, hydrocarbon

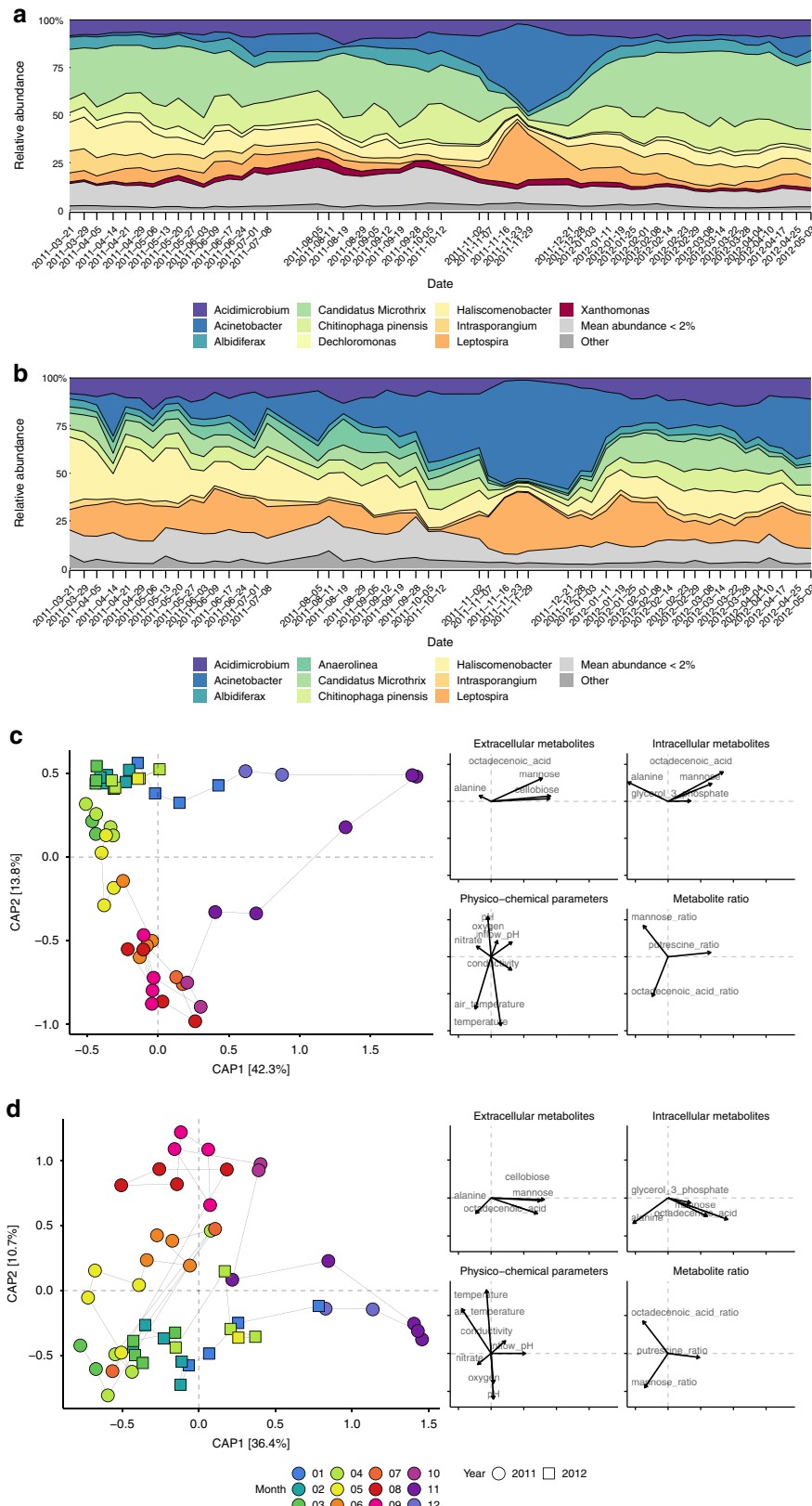

degradation, and TCA cycle, while fatty acid oxidation showed a marked peak. This highlights the transition from dominance by generalist, lipid-accumulating populations towards a lipolytic community.

With each of the four FunCs comprising multiple organisms encoding similar KOs and, hence, metabolic capabilities, we studied how individual populations adapt to their environment. To this end, we linked changes in community structure and in the expression levels of individual populations to the influence of environmental parameters. While rMAG abundance patterns could be linked to FunC assignment (Supplementary Fig. 3), we could not identify an analogous categorization when correlating

**Fig. 3 Community structure and function dynamics. a, b** Relative abundance and expression levels of recovered populations represented by rMAGs over time based on MG depth (**a**) and MT depth (**b**) of coverage, respectively, representing mapping percentages of MG [26% ± 3% (s.d.)] and MT [27% ± 3% (s.d.)]. The relative abundance of individual rMAGs is grouped based on genus-level taxonomic assignment with rMAGs of unresolved taxonomy grouped in "Other". Recovered genera with mean abundance below 2% are summarized as a single group (light gray). **c, d** Ordination of Bray–Curtis dissimilarity of relative abundances, MG (**c**) and MT (**d**), of individual rMAGs constrained by selected abiotic factors (metabolite levels, metabolite-ratios, and physico-chemical parameters shown as black arrows with arrow lengths indicating environmental scores as predictors for each factor). Points are colored by month of sampling and point-shape reflects the year of sampling. Thin black lines connecting the points visualize the time course of sampling. Source data are provided as a Source Data file.

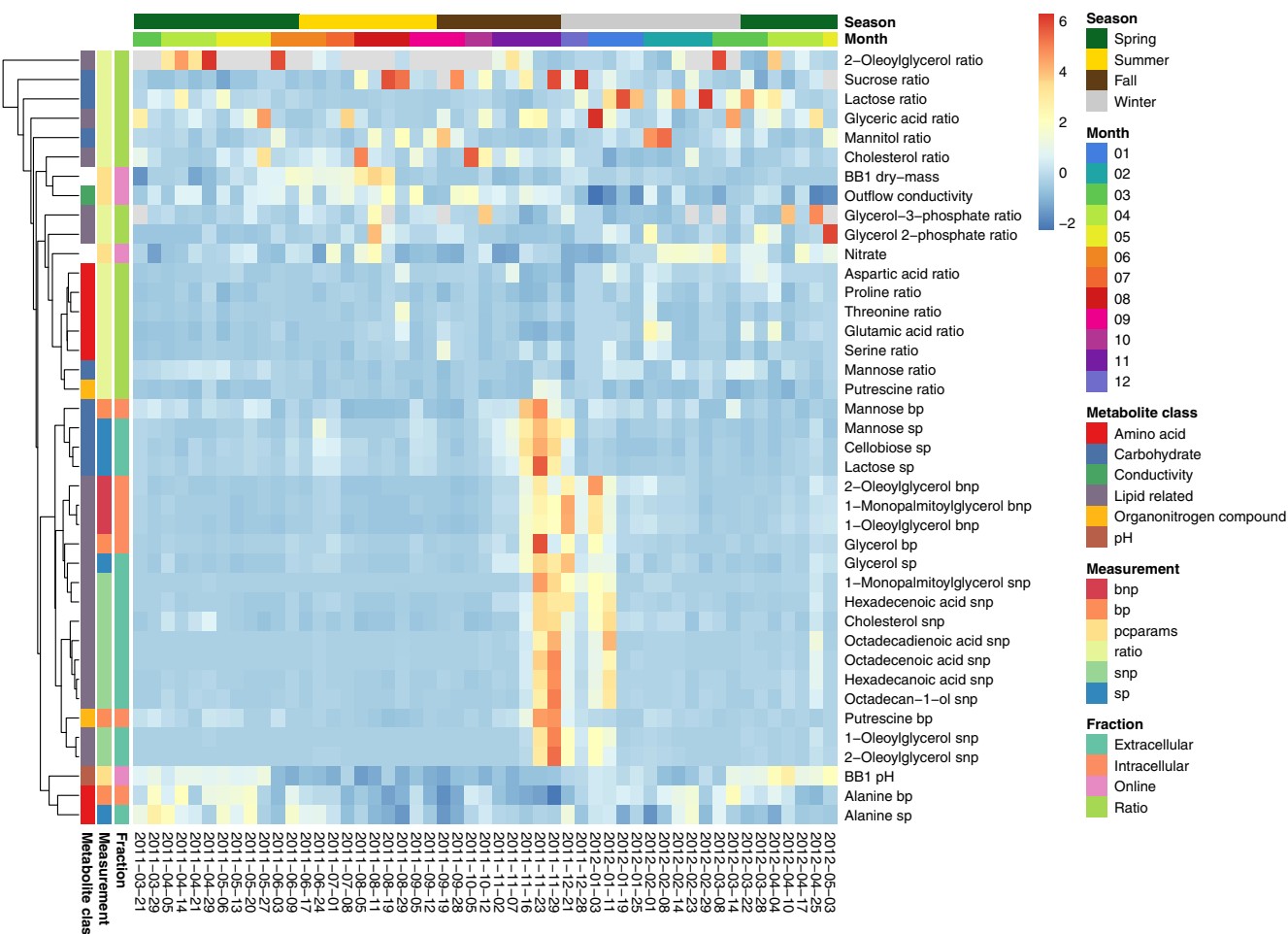

**Fig. 4 Levels of metabolites and physico-chemical parameters.** *Z*-score transformed metabolite intensities, metabolite ratios, and physico-chemical parameter levels over time are shown. Row annotations highlight classes of metabolites and parameters, measurement types (bnp: intracellular nonpolar metabolites, bp: intracellular polar m., pcparams: physico-chemical parameters, ratio: metabolite intrac./extrac. ratio, snp: extracellular nonpolar m., sp: extracellular polar m.), and the subtype or fraction of the measurement (manual: measured during sampling, online: measured during WWTP operation). Selected rows are shown (comprehensive heatmap shown in Supplementary Fig. 6). Source data are provided as a Source Data file.

rMAG abundances to abiotic factors. Instead, correlation patterns indicating similar preferences to environmental conditions emerged for subgroups of rMAGs across different FunCs (Supplementary Fig. 7). This shows that populations with a similar fundamental niche type responded differently to the environmental conditions pointing towards functional plasticity and, thus, adaptations of their realized niches

**Niche characteristics of in situ and ex situ time-series.** While we identified four fundamental niche types, it may be assumed that cohabiting species cannot occupy the same realized niches, leading to realized niche segregation within and between types. We hypothesized that different degrees of niche overlap, leading to variable levels of competition, must exist[40,41]. To better

understand the complementarity of realized niches, we used the functional omics data to study how rMAGs overlapped in relation to their encoded genes and how rMAGs varied in their expression profiles. While the former represents competition between populations with overlapping profiles, the latter is an important factor for the adaptability and overall survival strategy of individual populations. We distinguished between expressed KOs and nonexpressed KOs based on MT/MG ratios as well as MP data and computed distances between the resulting time-point-specific expression profiles. While the separation based on the functional potential was preserved in a clustering of expression profiles (in particular for FunC-2), the expression profiles of FunCs-1, FunCs-3, and FunCs-4 overlapped to a greater extent than those of FunC-2 (Fig. 6a). Two *Anaerolinea* populations assigned to

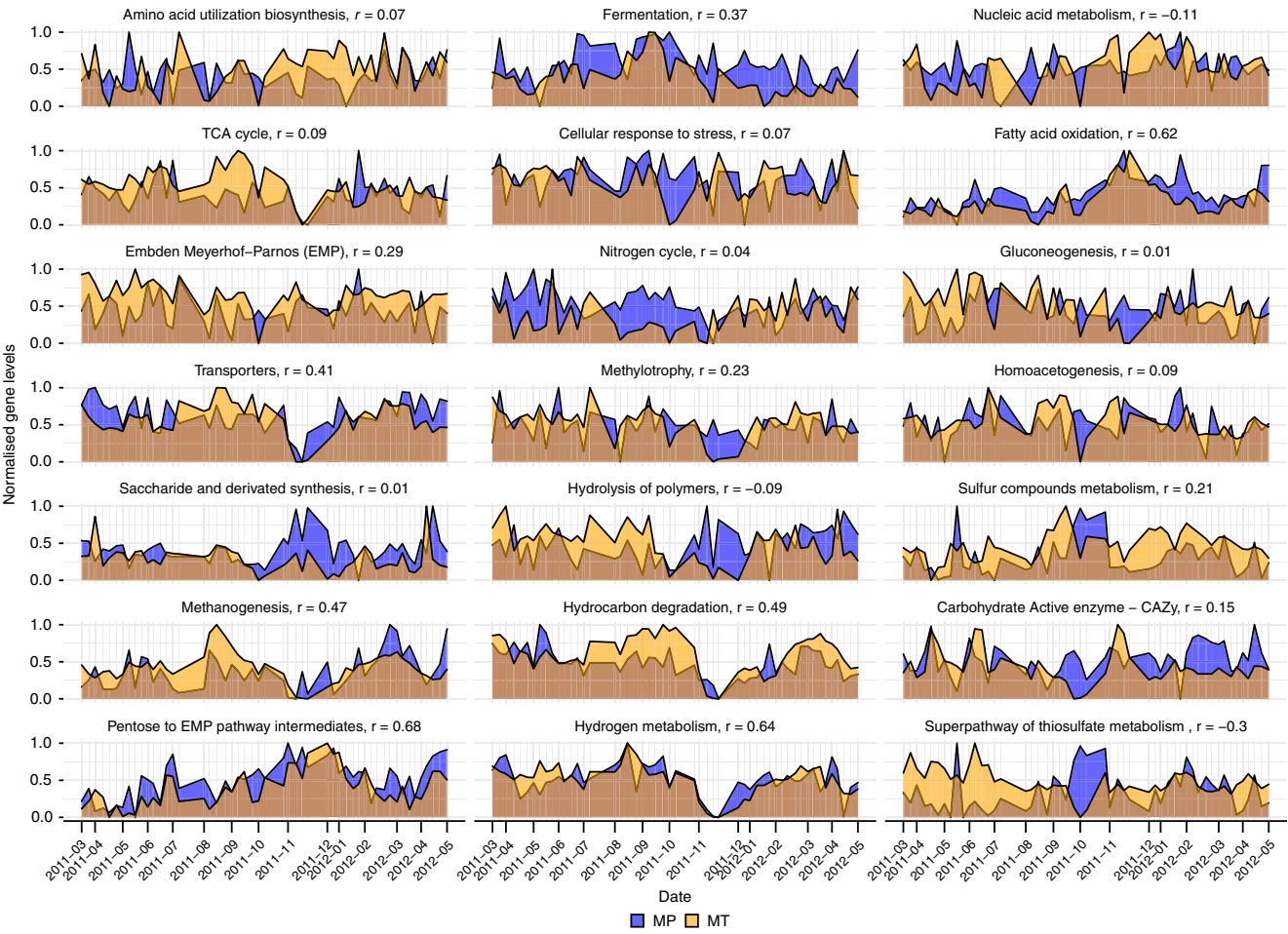

**Fig. 5 Gene levels over time grouped by functional categories.** Metatranscriptomic and metaproteomic levels (normalized relative expression for MT data and normalized relative spectral counts for MP data) of rMAG-derived genes assigned to FOAM ontology-based functional categories. Pearson correlation coefficients ($r$) of MT and MP values are shown in the title of each panel. Panels are ordered from highest to lowest mean MP relative count in row-major order. Source data are provided as a Source Data file.

FunC-1 appeared to express similar functions compared to the rMAGs of FunC-3 and FunC-4 and were found in a subgroup of rMAGs that showed a higher overall activity in terms of MT/MG ratios also when clustering expression profiles per time-point (Supplementary Fig. 8). Overall, the clusters based on KO expression status per time-point did not exhibit a separation according to the grouping into FunCs (Supplementary Fig. 8). This indicates a propensity of the respective rMAGs to more frequently express shared KOs than discriminatory KOs and, consequently, increased the competition for specific substrates.

To investigate the importance of individual, discriminatory functions, we selected rMAG clusters, based on gene expression and MP counts, to which the two most abundant rMAGs (D51_G1.1.2, A01_O1.2.4) had been assigned. We observed that clusters into which rMAG D51_G1.1.2 (*Microthrix*) was consistently categorized showed expression of few KOs with the majority being ribosomal proteins, TCA cycle-related enzymes such as pyruvate, malate, and glyceraldehyde 3-phosphate dehydrogenases, chaperones, and most frequently the WhiB family transcriptional regulator (19 time-points; Supplementary Data 6).

Clusters containing rMAG A01_O1.2.4 (*Acinetobacter*) frequently exhibited expression of genes related to motility and chemotaxis as well as stress response, but also functions related to phosphate accumulation, such as K08311 and K00937 (Supplementary Data 6). KOs related to lipid metabolism were also

frequently expressed in these clusters e.g. acylglycerol lipase (in 35 time-points) or diacylglycerol O-acyltransferase (25 time-points). This indicates that high expression of key functionalities is an integral part of the strategies of the populations within these clusters even though they differed with respect to their encoded functions.

We next studied how the observed distinction between populations with high activity is linked to phenotypic plasticity. As alternating oxygen levels in BWWTPs play an important role in selecting for lipid accumulating populations[7,42], we added oleic acid, the preferred carbon source for *Microthrix*[43], in lab-scale experiments under different oxygen fluctuation conditions[8] (see "Methods" section; Fig. 1). These ex situ conditions involved aerobic, anoxic, aerobically preconditioned biomass followed by hourly anoxic alternations, and anoxically preconditioned followed by hourly aerobic alternations. The MT/MG ratios for FunC-1 and FunC-3 were higher ex situ when compared to the in situ samples, and vice versa for FunC-2 and FunC-4 rMAGs (Fig. 6b). Furthermore, especially for FunC-3, average MT/MG ratios were highest in the aerobic conditions and lowest in the anaerobic conditions. This is in line with FunC-3 being comprised mainly of Betaproteobacteria and Gammaproteobacteria, which include mostly aerobic genera[44]. A more fine-grained view on differences in specific activity was obtained, when grouping rMAGs based on taxonomic assignment (Fig. 6c). While rMAGs of the classes Acidimicrobia and Actinobacteria (FunC-1)

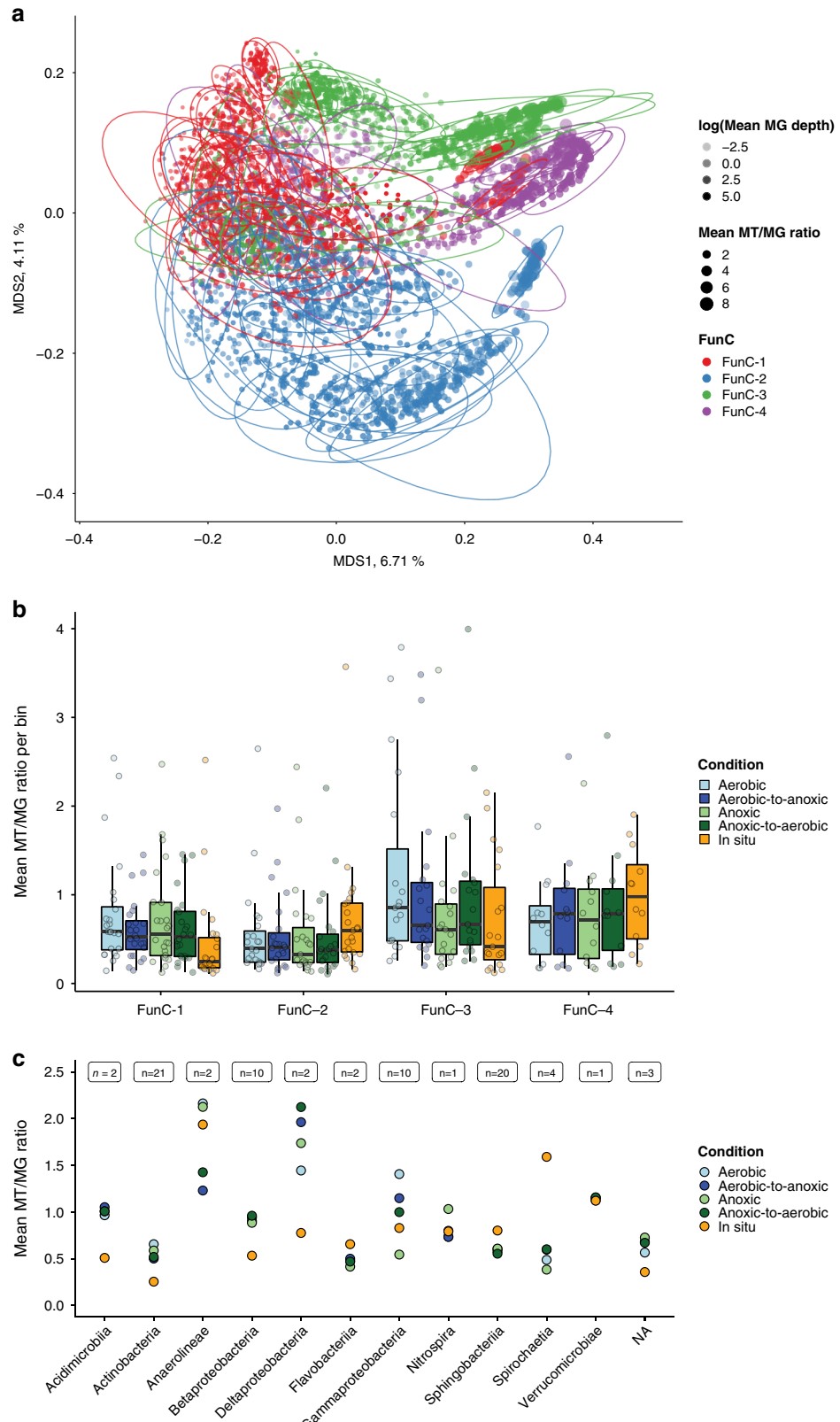

showed the lowest mean MT/MG ratios across the in situ time-series (0.5), the ratio was twice as high in the ex situ experiments across all conditions which can be attributed to the oleic acid pulse. Betaproteobacteria (FunC-3) behaved similarly, while Gammaproteobacteria (FunC-3) showed a tendency towards higher activity with increased oxygen levels. We observed

high activity for rMAGs assigned to Anaerolineae and Spirochaetia in the in situ time-series. Interestingly, this was not the case for Spirochaetia in the ex situ experiments, which points towards the necessity for additional substrates. The Anaerolineae rMAGs, with taxonomically related species being mainly anaerobic[45], showed the lowest MT/MG ratio under the

**Fig. 6 Realized niches. a** MDS of time-point specific expression profiles based on MT/MG ratios or evidence at the MP level. Colors indicate FunC assignment of the individual rMAGs. Point shape represents cluster assignment based on automated clustering of the embedded points. Ellipses containing 95% of cluster-assigned data points are shown. Points size represents the average MT/MG depth ratios of the individual rMAGs. The amounts of variance explained by the first two dimensions are shown on the respective axes. **b** Mean MT/MG depth ratios over all time-points are shown per condition for 78 rMAGs (boxplots show: center line, median; box limits, upper and lower quartiles; whiskers, 1.5× interquartile range; Each group of boxplots corresponds to a group of rMAGs (FunC-1 $n = 24$, FunC-2 $n = 23$, FunC-3 $n = 19$, FunC-4 $n = 12$), each boxplot represents an independent experiment.). **c** Mean MT/MG depth ratios grouped according to class-level taxonomic assignment of the rMAGs with the number of rMAGs for each group are shown in the top of the plot ($n$). Source data are provided as a Source Data file.

alternating conditions, while Deltaproteobacteria rMAGs showed high MT/MG ratios. Overall, the differentiated responses under alternating conditions point to distinct short-term and long-term adaptation strategies.

To study how fast the adaptations in response to the influx of oleic acid occur, we compared the baseline (0 h time-points, before oleic acid addition) against the 5 and 8 h time-points (after oleic acid addition). At 5 h, lipases, involved in TAG hydrolysis, for which high expression in the in situ samples was observed, were downregulated in the ex situ response to the addition of oleic acid (Supplementary Fig. 9a). An increased number of genes related to beta-oxidation were upregulated at 5 h, particularly in rMAGs assigned to FunC-3 (Supplementary Fig. 9b). Similar effects were observed when comparing the 0 h and 8 h time-points (Supplementary Fig. 10a, b). This suggests that responses in gene expression happen within the 5 h timeframe but on distinct time scales for different populations. In-depth analyses of the populations exhibiting the highest expression levels for TAG lipases, DGAT/WS, and PHB synthases (Supplementary Note 1 and Supplementary Figs. 11–14) underline the previously determined role of *Microthrix* as a key lipid accumulator in BWWTPs[13,46]. The results also indicate that populations such as *Anaerolinea*, *Leptospira* and *Acinetobacter* overlap with *Microthrix* in terms of their capacity to assimilate LCFAs and available neutral lipids. Niche complementarity and plasticity, i.e., overlapping fundamental and realized niches, as well as gene expression variability, impart population-independent processing of lipids in situ. From an ecosystem perspective, this community-wide trait confers functional resistance and resilience.

## Discussion

The ability to reconstruct population-level genomes and infer their functional potential from metagenomes allows identification of the fundamental niches of distinct community members. Unprecedented views of realized niches are achieved by tracking functional gene expression via MT and MP analyses, as well as actual resource usage resolved via comparative metabolomics analyses of intracellular and extracellular metabolites. The joint resolution of fundamental and realized niche breadths of individual populations is key to understanding the ecological processes within microbial communities, including, but not limited to, how such consortia respond to disturbance.

Here, through the application of our novel framework for the integration of multi-meta-omics datasets, we were able to track community-wide and population-resolved traits longitudinally in situ as well as ex situ. We found four distinct fundamental niche types in this ecosystem. Populations assigned to a specific type shared common functional repertoires and largely shared a similar phylogenetic background, in line with previously observed metabolic repertoires[47,48]. Simultaneously, some functions, e.g., related to lipid accumulation, were found to be enriched in multiple niche types.

Despite our results showing a link between functional complement, realized niches, and phylogeny, we also observed distinct activities in response to the changing environmental conditions

within individual niche types, e.g., some lowly abundant populations exhibited high activity. This suggests distinct adaptation strategies to variabilities in the resource space and is exemplified by the populations in the functional cluster that includes the dominant *Microthrix* population. *Microthrix* follows a strategy based on phenotypic heterogeneity for rapid adaptation to the prevailing environmental conditions[13]. Our ex situ validation experiments revealed the adaptations to changes in substrate availability and dissolved oxygen concentrations after as little as 5 h post-disturbance. This plasticity in gene expression allows the populations to be resistant to fluctuations in environmental conditions. Furthermore, this strategy was found to be unique to *Microthrix* as evidenced from the increased transcriptional response of other lipid-accumulating and/or lypolytic populations, e.g., *Acinetobacter*, *Leptospira*, or *Anaerolinea* spp., especially in the aerobic ex situ conditions.

Our work highlights the requirement to account for organism-specific adaptation strategies and time-frames within mixed communities. We observed that drastically altered community composition and gene expression patterns followed a severe disturbance in substrate levels within our time-series. We hypothesize that this community shift was a consequence of excess substrate availability, and it highlights a limit to the community's resistance. Individual populations recovered within ten sludge age cycles post-disturbance, which indicates that the resilience of the community is also linked to phenotypic plasticity. The overlap in realized niches reflects niche complementarity. This in turn is governed by interspecific competition over a set of substrates, such as oleic acid. Other independent work on the human gut microbiome has highlighted the importance of interspecific competition for the maintenance of stability under a constant feeding regimen[49]. How interspecific competition or lack thereof relates to resilience represents a key question for future work.

Overall, our framework demonstrates that multi-meta-omics data allows an in-depth characterization of ecological niches over time. Due to the observed plasticity in activity and the recovery after a major, transient perturbation, we confirm that the relationship between resistance and resilience is a function of fine-scale competition over resources in this environment. The resulting complementarity in both the fundamental and realized niches guarantees the provision of stable ecosystem services[50] and, thus, the long-term stable operation of mixed-culture biotechnological processes. These results are particularly relevant for the future engineering of niches within mixed-culture biotechnological processes[3], which are key to achieve humankind's sustainability goals[1,2]. In more general terms, it will be important to understand if phenotypic heterogeneity and niche complementarity play similarly important roles in the stability of other microbiomes.

## Methods

**Sampling and biomolecular extractions**. Oleaginous biomass comprised of floating sludge islets was sampled from the surface of an anoxic tank at the Schifflange municipal biological wastewater treatment plant (BWWTP; Schifflange, Luxembourg; 49°30′48.29″N; 6°1′4.53″E)[27]. In situ sampling intervals of approximately one week were chosen to match the sludge age (the average time the

biomass remains in the entire system) as well as the average doubling time of the dominant *Microthrix* population[7,13].

Samples were collected with a levy cane, stored in 50 mL sterile Falcon tubes and flash-frozen on site. Biomolecules were extracted in randomized batches after the end of the sampling period. A total of 53 samples was extracted. The set included two preliminary samples (2010-10-04 and 2011-01-25) and 51 samples from a higher frequency sampling phase (2011-03-21 to 2012-05-03).

Polar and nonpolar metabolites, DNA, RNA, and proteins were extracted in a sequential co-isolation procedure[13,51]. Around 200 mg of frozen samples were weighed out. Extracellular metabolites were extracted from the supernatant with cold chloroform and methanol–water, and separated into polar and nonpolar fractions. Intracellular metabolites were isolated in the same way after a lysis step by cryomilling, followed by sequential spin column-based (Qiagen Allprep) purification of RNA, DNA, and proteins.

**Abiotic factor measurements and data processing**. At the time of sample collection, the following physico-chemical parameters were measured inside the tank with a portable field kit (Hach) on-site: pH, conductivity, oxygen-levels, and temperature.

Additionally, online monitoring measurements were recorded by the BWWTP operators including nitrate, phosphate, ammonium, dry-matter and dissolved oxygen levels at the outflow as well as conductivity and pH at the inlet, and pH and temperature inside the sampled tank (referred to as operational measurements). Six missing values in the on-site measurements for pH were imputed from the available measurements with the R-package imputeTS using the method *stine*[52].

**Metaproteomic analyses**. Protein samples were separated by 1D SDS-PAGE (Criterion precast 1D gel, Bio-Rad), stained and cut into 2 mm bands. Peptides were subjected to liquid chromatography (LC) after in-gel reduction, alkylation and tryptic digestion. An Easy-nLC column (Proxeon, Thermo Fisher Scientific) was used. The peptide mixture was separated with a binary solvent gradient for elution with 0.1% formic acid in water and 0.1% formic acid in acetonitrile. Mass spectrometry was performed with an LTQ-Orbitrap Elite (ThermoFisher Scientific) on an 11-scan cycle consisting of a single precursor scan at a mass range of 300−2000 *m/z* followed by ten data-dependent MS/MS scan events. MS/MS scans were carried out with an isolate width of 2 *m/z* and a normalized collision energy of 35. Additional details of the metaproteome preparations and measurements are described in a previous study[13].

We converted raw mass spectrometry files to MGF format using MSconvert[53] using default parameters. The Graph2Pro pipeline[54] was used to process the resulting files together with the corresponding MG and MT co-assembly graphs from MEGAHIT[55]. The Graph2Pro pipeline uses FragGeneScan[56] to predict proteins from the long edges in the assembly graphs (i.e., from the contigs). In addition, it predicts tryptic peptides that span multiple edges in the graphs. Search databases were constructed using the putative proteins and tryptic peptides, respectively. These were used for initial peptide identification with the MS/GF+ search engine[57]. Identified tryptic peptides were then combined and used as the constraints for Graph2Pro to predict protein sequences from the co-assembly graphs. The generated sample-specific databases produced by Graph2Pro were used for the final metaproteomic searches using MS/GF+ (second search) to produce the final identification results. MS−GF+ was used for the final peptide identification with custom parameters: the instrument type was set to high-resolution linear trap quadrupole (LTQ) with a precursor mass tolerance of 15 ppm and an isotope error range of −1 and 2 and the minimum and the maximum precursor charges were set to 1 and 7, respectively. We estimated the false discovery rate (FDR) with a target-decoy search approach using reverse sequences of the protein entries while preserving the C-terminal residues (KR). An FDR threshold of 1% was used. Identified peptides from the Graph2Pro pipeline were assigned to coding sequences (CDS) of rMAGs from prokka-based[58] predictions (see below) by using the command line interface version of peptidematch[59]. Spectral counts for sample-specific peptide sequences were assigned to matching CDS.

**Meta-metabolomic analyses**. Four distinct measurements for the metabolite extracts were performed: i) nonpolar extracellular, ii) polar extracellular, iii) nonpolar intracellular, and iv) polar intracellular. Metabolite extracts were derivatized using a multipurpose sampler (GERSTEL). Dried polar samples were dissolved in 15 μL pyridine, containing 20 mg/mL methoxyamine hydrochloride (Sigma-Aldrich), and incubated under shaking for 60 min at 40 °C. After adding 15 μL of N-methyl-N-trimethylsilyl-trifluoroacetamide (MSTFA; Macherey-Nagel), samples were incubated for additional 30 min at 40 °C under continuous shaking. Dried nonpolar samples were dissolved in 30 μL MSTFA and incubated under shaking for 60 min at 40 °C. For quality control, pool samples, i.e., a combination of all extracts of the same measurement[27], were introduced in the measurement sequence after every fifth measurement.

GC-MS analysis was performed using an Agilent 7890A GC coupled to an Agilent 5975C inert XL Mass Selective Detector (Agilent Technologies). A sample volume of 1 μL was injected into a split/splitless inlet, operating in splitless mode (intracellular and extracellular polar fraction) and split mode (10:1, intracellular non-polar fraction) at 270 °C. The gas chromatograph was equipped with a 30 m (I.

D. 250 μm, film 0.25 μm) DB-5MS capillary column (Agilent J & W GC Column). Helium was used as carrier gas with a constant flow rate of 1.2 mL/min.

The GC oven temperature was held at 80 °C for 1 min and increased to 320 °C at 15 °C/min. Then, the temperature was held for 8 min. The total run time was 25 min. The transfer line temperature was at a constant 280 °C. The mass selective detector (MSD) was operating under electron ionization at 70 eV. The MS source was held at 230 °C and the quadrupole at 150 °C. Full scan mass spectra were acquired from *m/z* 70 to 700.

All GC–MS chromatograms were processed using the MetaboliteDetector software[60] (v. 2.5). The software package supports automatic deconvolution of all mass spectra. The following deconvolution settings were applied: peak threshold: 6, minimum peak height: 6, bins per scan: 10, deconvolution width: 2 scans, no baseline adjustment, Minimum 15 peaks per spectrum, No minimum required base peak intensity. Compounds were automatically annotated by retention time and mass spectrum using an in-house mass spectral library. Detected metabolite derivatives (_*x*MeOX_*x*TMS/_*x*TMS) were used for further statistical data analysis.

Metabolites detected in blanks at a mean intensity level of more than 75% of the mean level in samples were removed as contaminants. Metabolites that were not detected in all pool samples were also removed from subsequent analysis as well as metabolites not detected in at least 25% (90% for correlation analyses) of samples. Metabolite intensities were normalized with respect to pool samples to account for instrument drift as described previously[51] by dividing the intensity values by the mean of up to two preceding and subsequent pool samples according to the measurement sequence. Metabolite derivate names of identified metabolites were manually assigned to KEGG compound identifiers and CHEBI IDs.

**Metagenomic and metatranscriptomic analyses**. MG libraries were prepared as paired-end libraries with the AMPure XP/Size Select Buffer Protocol following a size selection step[13]. RNA libraries were prepared after washing stored extracts with ethanol and depletion of rRNAs with the Ribo-Zero Meta-Bacteria rRNA Removal Kit (Epicenter). The ScriptSeq v2 RNA-Seq Library Preparation Kit (Epicenter) was used for cDNA library preparation. Libraries were sequenced on an Illumina Genome Analyser (GA) IIx instrument with a read-length of 100 bps paired-end. Downstream processing and assembly of MT and MG reads was carried out with IMP[28] version 1.3 with the following parameters: i) Illumina Truseq2 adapters were trimmed, ii) the filtering step for reads of human origin was omitted, and iii) the MEGAHIT v.1.0.6 de novo assembler[55] was selected for coassembly of the MG and MT data. Co-assembled contigs from each timepoint were binned based on nucleotide signatures, presence of single-copy essential genes and metagenomic depth of coverage[61]. MAGs from each timepoint with at least 28% completeness and with a contamination of less than 20% based on essential marker gene content[62] were retained for downstream selection of representative population-level genomes (rMAGs). To this end, MAGs were dereplicated with dRep[29] using the following parameters: i) completeness threshold of 0.6, ii) strain heterogeneity threshold of 101, iii) primary cluster identity of 0.6, and iv) secondary cluster nucleotide identity of 0.965, and other parameters at default settings. In a following step, a subset of rMAGs with the highest completeness rates was selected based on CheckM[63] completeness estimates, requiring at least 0.50 in the difference of completeness and contamination estimates. Furthermore, rMAGs without taxonomic assignment on kingdom level were removed as they could represent misassembled contigs, resulting in a set of 78 coherently taxonomically annotated rMAGs that were used for the time-series analysis. For downstream analyses, MG and MT reads from all time-points were mapped using bwa mem[64] per time-point using the rMAGs as references. MG and MT depth-of-coverage per time-point were computed on the gene and contig level by dividing the summed depth per base by the length of the respective sequence.

Assembled contigs from IMP were annotated with Prokka v1.11[58] including prediction of full-length coding sequences (CDS) with prodigal v2.60[65]. Predicted CDS were also searched with an in-house Hidden Markov Model (HMM) database[61] of KEGG ortholog groups (KO) using HMMer v.1.12b[66]. We inferred compounds linked to CDS through CDS-to-reactions links from predicted enzymes with their respective KO annotation and EC assignment. Links to FOAM ontology categories[67] were assigned to each CDS by matching KO annotations. To assign MIMAG classifications[68] for all MAGs, assembly statistics, e.g., N50, were computed with the R-package seqinr v3.6-1[69]. tRNAs were predicted with Aragorn v1.2.38[70] and MAGs were screened for rRNA genes with barrnap 0.9[71].

Taxonomic assignment of rMAGs was performed using AMPHORA2[72] in combination with sourmash-lca v. 2.0.0a1[73], kmer-length:21 and threshold:4 and an existing database including approximately 87,000 microbial genomes (downloaded on 2017-11-09 from https://osf.io/s3jx8/download). If no taxonomic assignment was possible by whole genome-comparison (sourmash-lca), predictions for unassigned levels were augmented with consensus predictions using AMPHORA2: Assignments based on individual marker genes were combined by summation of the associated assignment probabilities. The consensus assignment with the highest overall score was determined. If the consensus assignment scores constituted for less than one third of the total probability scores the assignment was discarded as "low confidence assignment".

**Ex situ experiments**. The ex situ experiments were performed in bioreactors seeded with sludge samples and diluted 1:5 (v/v) with artificial wastewater with a

final volume of 2 l[8]. The mixed sludge was split into two aliquots subjected to aerobic or anoxic preconditioning for 2 h. For the anoxic preparations a gaseous dioxygen-free environment (<1000 ppm) was achieved and monitored within a glove box (Jacomex, Dagneux, France). Thirty milliliters of the sludge mixtures were transferred into 50 mL serum vials connected to a multifold valve system to generate alternating aerobic (compressed air) or anaerobic (nitrogen gas) conditions. Samples were subjected to aerobic, anoxic, or alternating conditions (in 1 h intervals) after 2 h of preconditioning. After the preconditioning (time-point 0 h), oleic acid was supplemented at 500 μM alongside nitrate (80 μM) and phosphate (16 μM). Additional samples for concomitant DNA and RNA extraction and sequencing were taken at 5 and 8 h. This resulted in 12 samples for the four conditions tested (aerobic, anoxic, aerobically preconditioned followed by alternating, and anoxically preconditioned followed by alternating).

Isolated DNA for the 12 samples was sequenced on an Illumina Hiseq 2500 with a read-length of 250 bps paired-end. Isolated RNA was reverse transcribed to cDNA and sequenced with a read-length of 100 bp paired-end. Resulting MT and MG reads were pre-processed with IMP[28] and mapped to the rMAGs reconstructed from the long-term time-series as described above. Raw read counts per CDS were determined with featureCounts[74] and compared with DESeq2[75] across all conditions.

**Statistical analyses**. Statistical analyses were performed using R 3.4.4 and R 3.6.1[76] with prevalent use of the tidyverse R-package[77].

**Determination of niche types**. Annotated KOs for the individual rMAGs were summarized in a binary presence/absence matrix, in which 0 s indicated absence and 1 s indicated presence of at least one gene annotated with the respective KO. Subsequently, pairwise binary Jaccard-distances between the rMAGs based on these KO profiles were calculated and projected into two-dimensional space by multidimensional scaling (MDS). To determine the clustering of rMAGs in the resulting embedding, the appropriate number of clusters was determined by utilizing the k-means function for a range of clusters (ranging from 1 to 9 centroids) and determining the total within-sum-of-squares error as a measure of variability of resulting clusters. Functional clusters (FunCs) were then determined by k-means clustering. Enrichment of individual KOs within the assigned FunCs was determined with Fisher′s exact test based on the number of rMAGs with the assigned KO. Resulting p-values were adjusted by FDR correction (function p.adjust method = "fdr") and KOs with a p-value below 0.05 were considered as enriched within a FunC. To test the relationship of rMAGs abundance and FunC assignment, pairwise Pearson correlation (cor.test in R) was computed between the relative abundance values of the rMAGs across time. Resulting correlation coefficients ($\rho$) were transformed to distances with the following formula: $1 - \frac{\rho+1}{2}$. Dispersion of these distances was assessed with the betadisper function of the vegan package[78]. Association of the FunC assignment to the distances was tested using the adonis function.

Whole genome-based pairwise distances between all rMAGs were calculated with mash v.2.2.2[30] (-k 21 −s 10,000) and embedded in two dimensions using MDS. PROCUSTES from the vegan package[78] was used to map the whole genome-based embedding onto the KO profile-based embedding. PROTEST from the vegan package was used with 9999 permutations.

**Linking abiotic factors to population abundances**. Measurements of abiotic factors (metabolites and physico-chemical parameters), as well as ratios of intracellular and extracellular metabolite intensity ratios were transformed to z-scores. Relative abundances of rMAGs were associated to abiotic factor levels by testing for correlation (cor.test function, method = "spearman"). Additionally, abiotic factor levels were placed onto a 2D ordination of MG or MT-based abundance profiles (Bray–Curtis dissimilarity) applying the vegan function scores.

**Correlation of gene levels**. To assess the expression of pathways over time, MT depth and MP spectral counts for genes of rMAGs were summed for each L1 FOAM category[67]. Grouped values were divided by the total MT depth or MP counts of all rMAGs per time-point to obtain the relative contribution of genes assigned to a specific category. The relative values were scaled to values between 0 and 1. Correlations between the scaled MT and MP time-series for each functional category were calculated with the cor function in R.

**Clustering of expression profiles**. To characterize expression profiles of the distinct rMAGs over time, gene functions (KOs) were summarized as active or inactive depending on MT/MG ratios or evidence at the MP level. KOs were considered active if at least one gene with the KO matched the following criteria: either the MT/MG depth ratio of the gene was greater than 1 or at least 2 peptide spectral counts could be assigned to the gene. If the MG depth of a gene was below one, the MT depth was considered instead of the MT/MG ratio to avoid inflating active KOs for lowly abundant populations. Binary Jaccard distances of the resulting KO profiles were determined for each rMAG separately for each time-point. Clusters were determined in each of the resulting 51 ordinations with the hdbscan function of the dbscan package with a minimum of five members per

cluster. The resulting clusters were used to assess which functions were expressed over time by different subsets of rMAGs with a focus on rMAGs assigned to the same clusters as Microthrix D51_G1.1.2 or Acinetobacter A01_O1.2.4.

**Reporting summary**. Further information on research design is available in the Nature Research Reporting Summary linked to this article.

## Data availability

Meta-omics data from five individual time-points has previously been published[13,26,51]. The MG and MT FASTQ files and the sample-wise MT-assembly and co-assembly contigs are available on NCBI BioProject PRJNA230567. MP data has been deposited in the PRIDE database under the accession number PXD013655. Raw metabolomics data is available at MetaboLights under the accession MTBLS2021, while processed intensities after identification are provided with this manuscript (Supplementary Data 3). Similarly, physico-chemical parameters are provided with this manuscript (Supplementary Data 4). Processed and intermediary data files from the combined multi-omic analyses, e.g., annotated and normalized MT, MG read counts, are available at Zenodo (https://doi.org/10.5281/zenodo.3961685). External databases were used in this study: KEGG (https://www.genome.jp/kegg/), CHEBI (https://www.ebi.ac.uk/chebi/).

## Code availability

Code used for genome reconstruction and dereplication is available at the LCSB R3 GitLab (https://git-r3lab.uni.lu/shaman.narayanasamy/LAO-time-series). Code used for the processing and analyses of the meta-omics data, as well as for additional analyses and generation of plots for main and supplemental figures is also available at the LCSB R3 GitLab (https://git-r3lab.uni.lu/malte.herold/laots_niche_ecology_analysis).

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

## Acknowledgements

We thank Mr Bissen and Mr Di Pentima from the Syndicat Intercommunal a Vocation Ecologique (SIVEC), for their permission to collect samples and access to the monitoring platform of the Schifflange wastewater treatment plant. We also thank Dr. Olivia Judson for her feedback on the title and abstract. Bioinformatic analyses presented in this paper were carried out using the high-performance computing facilities of the University of the Luxembourg. This work was supported by an ATTRACT programme grant (ATTRACT/A09/03) and INTER programme grant (INTER/SYSAPP/14/05), two CORE programme grants (C15/SR/10404839, C17/SR/11689322), and a PRIDE doctoral training unit grant (PRIDE15/10907093) to P.W., and a grant to A.R.S (PDR-2013-1/5748561) all funded by the Luxembourg National Research Fund (FNR). As one of the authors (A.R.S.) holds a position at the European Commission the following applies: "The views expressed are purely those of the writer and may not in any circumstances be regarded as stating an official position of the European Commission."

## Author contributions

E.E.L.M., L.A.L., H.R., A.R.S. and P.W. performed the sampling and the biomolecular extractions. M.H., S.M.A., S.N., A.H.B., P.M., B.K., C.C.L., and P.W. analysed the data. A.R.S. performed short-term experiments and biomolecular extractions. I.B. and R.B.H.W. performed RNA and DNA sequencing for the ex situ experiment samples. J.D.G., J.M.S., and P.S.K. performed RNA and DNA sequencing for the in situ time-series samples. C.J. performed the metabolomics experiments and data analysis. M.R.H and R.L.M. measured the proteomic data. M.H., B.J.K., Y.Y., S.L., and H.T. analysed the proteomic data. M.H., E.E.L.M., and L.A.K.K.B. performed the metabolic characterization of the reconstructed genomes. M.H., E.E.L.M., A.H.B., P.M., C.C.L., and P.W. designed the study and wrote the manuscript. All authors discussed the results and commented on the manuscript.

## Competing interests

The authors declare no competing interests.
