## [Peer Review File · Nature Communications]

REVIEWER COMMENTS

Reviewer #1 (Remarks to the Author):

The manuscript by Harald et al. describes the use of a multi-omics approach to determine the phylogenetic and functional shifts that occur in a mixed-culture biotechnological process. Samples were collected from a municipal wastewater treatment plant over a period of 14 months, allowing detailed understanding of community fluctuations over time and responses to disturbances. Additional ex situ experiments were conducted with oleic acid feeding under different oxic conditions to evaluate the short-term responses to pulse disturbances. The study resulted in a large number of metagenome assembled genomes (MAGs) which were used to define different ecological niches based on the biochemical potentials of the MAGs.

As an approach, this is an extremely elegant study that takes advantage of the current state-of-the-art in multi-omics technologies; approaches that can currently only be carried out by a few select laboratories, including that of the senior author.

Their work highlights the importance of interspecies niche complementarity for metabolism of complex substrates, such as encountered in wastewater plants. The main novel finding is that the MAGs could be divided into different fundamental niches. The resulting representative MAGs (rMAGs) were studied to determine their functional capacities based on their predicted gene functions. The temporal dynamics and activities of the rMAGs in the wastewater treatment plant over the 14-month period was determined based on read mapping of MGs and MTs, respectively. The study also distinguished between genes (KOs) that were expressed compared to those that were not expressed based on MT/MG ratios and proteomics. This enabled them to study the plasticity of gene expression during shifts in environmental conditions.

The rest of the findings were more descriptive in nature and it is difficult to judge what is novel over what is already known about community structure and dynamics in wastewater treatment plants. The dominant *Microthrix* organism has been extensively studied in previous research.

What about the metaproteomics results? Lines 123-125 mentions how many spectra were obtained and that they were matched to MAGs. Where are the results? The only mention of the proteome data in the results is on line 300, but I couldn't find any description of the results or a figure. The only proteomics data were found on Supplementary Figure 12 for the ex situ experiment and that figure is not mentioned in the text.

Note that Supplementary figures 9,10, 11 and 12 are not mentioned in the manuscript.

Minor comments

Assume that this is hypothesis generating research? I couldn't find a hypothesis that was tested.

By reducing the description of methods in the results section, it might be possible to add more results, for example from the proteomics data.

Fig. 3 figure legend doesn't mention the metabolite ratio graphs.

Figure 4 is labeled "abiotic factors" but is a heatmap of metabolites. Consider a better figure title and more complete figure legend.

Reviewer #2 (Remarks to the Author):

This is a powerhouse manuscript that provides an unprecedented level of molecular and physiological detail into a wastewater treatment microbial community. The authors apply a wide range of -omics-based methodologies combined with state-of-the-art bioinformatics in a time-series manner to track dynamics in community composition and functionality. The manuscript is incredibly well written, the results are clear, and the main conclusions appear well supported by the data. All together, I commend the authors on their enormous efforts. While I am deeply impressed by the incredible amount of work and effort that went into this manuscript, I have several serious concerns.

1) DESCRIPTIVE/EXPLORATORY

The study is very descriptive in nature. There is no clear question or hypothesis stated in the introduction, and the manuscript very much reads as an exploration of the meta-omics datasets. I am left wondering why the authors performed the study in the first place, as the disturbance event was likely fortuitous. The lack of a clear question or hypothesis is not a problem for a field-specific journal, such as ISME J, but I find it problematic for a broad audience journal where the reader may not be accustomed to such a highly descriptive study.

2) RATIONALE

The authors use biotechnology as the rationale for their study (lines 26-27, 44-46, 411-412). However, it is unclear to me how any of the main outcomes could be used in practice. How would the knowledge gained help us to design, improve, and/or optimise a biological process? This needs explained.

3) MECHANISTIC INSIGHT

As is the typical problem with descriptive studies, I am missing mechanistic insights. For example, the authors observe a correlation between changes in certain substrates and community dynamics. What is the direction of causation? Do substrate changes cause community changes or vice versa? What is the underlying cause of the substrate changes? I am not convinced that the ex situ experiments help to understand this, as the in situ and ex situ substrates are different. As with point one above, the lack of convincing mechanistic insights is not a problem for a field-specific journal, such as ISME J, but is somewhat unsatisfying for a broad audience journal.

4) GENERALITY

I assume that many of the observations are likely system specific. Which take-home messages are potentially generalisable and which are not? This needs discussed. If there are no clearly generalisable outcomes, then I again question whether this manuscript is suitable for a broad audience journal.

Reviewer #3 (Remarks to the Author):

Herold and colleagues present a well-designed experiment to identify how fundamental and realized niches deviate from one-another for microbes in sewage sludge. By comparing fundamental niches based on metagenomics to realized niches based on repeated metatranscriptomics and metaproteomics, the authors concluded that the genomic repertoire was a poor predictor of relative abundance in response to changes in the abiotic environment. The authors then did a follow-up study in which they incubated sludge under a range of oxygen conditions to assess microbial plasticity on shorter time scales. This is an impressive dataset and a very interesting framing of the research at hand. I am unfamiliar with other papers which have taken such a holistic approach to understanding the interaction between ecology and

biotechnology, and think this paper could be of broad interest to researchers in both these fields. I provide some comments below, primarily related to the role of phylogenetically-conserved core genes in occluding functional differences.

MAGs:

Please report the quality data for each MAG used in the paper (contamination, completeness, taxonomy, coverage across the MAG etc as in doi:10.1038/ismej.2015.241). Table S2 just has the relative abundance of the MAGs.

Also, I would imagine that the presence and absence of genes (ie niche assignment) would be contingent upon genome completeness (and also perhaps niche assignment might actually correlate with completeness for biological reasons such as differences in relative abundance under different biotic conditions). Did you try re-running the analysis either with resampling each genome to the same completeness, or accounting for the differences in completeness/contamination in some other way? [L738]

MAG clusters:

L151: If genomes cluster by taxonomy, how do you attribute differences in niche related to functional genes rather than some longer, phylogenetically-conserved ability to show similar responses to a changing environment? I think it is essential to differentiate between functions which are enriched for because certain areas of the phylogenetic tree are enriched for, versus functions which are enriched for independent of the genetic background or whole-organism evolutionary history. The authors do discuss the taxonomic contribution in place (ex. L190, 203), but do not account for it in their conclusions.

One way to address this is to ask how the genomes of Actinobacteria which end up in cluster FunC1 differ from the complete pool of publicly-available Actinobacteria genomes (ie remove the "core" Actinobacteria genome)? Does genomic potential within this group constrain it to existing only in FunC1? Or do other members of Actinobacteria have many of the functional genes required to belong in other clusters, but the phylogenetic signal of the core genome is overwhelming? For instance, I think Actinobacteria genomes have been found to have nirK but be unable to complete denitrification (<https://doi.org/10.1371/journal.pone.0114118>). So this seems like the taxonomic (or phylogenetic) and functional niches cannot be separated, and that these clusters are only interesting insofar as they consist of organisms which have traits that cannot just be explained by ancestry. Another way to remedy this would be to run a phylogenetic PCA and see if you get the same "functional" clusters in the MAGs. Then you can think about how phylogeny defines fundamental niche, but does not constrain the realized niche. For instance, is the phylogenetically-corrected functional gene distance matrix correlated with the environmental response dissimilarity matrix for the taxa?

Language:

I found the language was impenetrable in parts of the introduction. There are a lot of big words and complicated sentences, and it would be nice to provide the reader with some simpler language periodically so their mind can rest. The sentences between lines 70-75 were particularly challenging.

Methods:

For people (like me!) who aren't familiar with how these anaerobic wastewater treatment chambers work, it might be useful to quickly state what the microbial community retention time is/how this thing works. So please move the text on L 393 regarding the rationale for sampling interval up to the beginning of the results. Does this flow/timeframe make the system more like a retentostat or a chemostat? Basically, do you need to account for the metabolites/proteins in the inflow in your data, or are the majority of compounds produced in-situ [L227-229 measures metabolites as substrates in the mat, not the incoming substrates, correct)?

The MAGs allow for population-level genomes, so are the observed patterns likely to be cell phenotypic plasticity, or due to shifts in microbial ecotypes within a population? Could this be tested with approaches like the one in Bendall et al. (doi:10.1038/ismej.2015.241)? Or are the genomes not of sufficient quality? Or are you mapping the metaT reads to a single representative MAG from across all the timepoints already?

L769 – how do the authors have a metagenome depth less than 1? Is this because they co-assembled metaT and metaG data to get the MAGs, so the gene was expressed without the gene being in the metaG data? If so, does this mean the MAGs cannot be trusted because of low coverage?

I think your statistics about enriched genes need to be corrected for this lack of phylogenetic independence when comparing MAG and clusters of MAGs (L177-191).

Overall, this is a very nice paper. Good work!

Reviewed by Grace Pold

Reviewer #4 (Remarks to the Author):

Herold et al. present an extensive omics analyses to reveal how microbial ecosystems respond to disturbance. The analyses include metagenomics, metatranscriptomics, metaproteomics and metabolomics based on both in situ sampling and ex situ lab experiments. Although a lot of data analysis work was conducted and presented in the manuscript, I have strong concerns about the reliability of the analysis processes (especially metagenomic analyses) and the key findings of the study.

Main concerns:

1. It's not very clear what are the key findings of this study. From the abstract section, I mainly got two main messages: (1) the authors uncovered four distinct fundamental niche types; (2) The change of community structure coincided with the changes of substrate availability. For the first one, it's not difficult to identify some different clusters/types from bioreactor bacterial community. After reading the manuscript, I'm not quite clear what is the significance of the four types identified in this study? How do they relate to the main functions (transformation or removal of organics, nitrogen and phosphorus) of the wastewater treatment bioreactors? As for the second message, I think it is a common sense for researchers and engineers in the field of wastewater treatment.

2. It seems that the metagenomic analysis has serious flaws. (1) For the MAG construction, the authors applied a very coarse and arbitrary quality filtration criteria (>28% completeness, <20% contamination). To my knowledge, this is not acceptable at all. Such a low completeness and high contamination will dramatically affect the reliability of the down-stream analysis. The minimum required information about a metagenome-assembled genome of bacteria and archaea can be found in this paper <https://www.nature.com/articles/nbt.3893> and many other published papers. (2) Another concern about the metagenomic analysis is that the authors used many low-quality MAGs to investigate the bacterial community. To me, this is also not acceptable for the following two reasons. First, currently, there are no reliable methods to assign a MAG accurately to a low taxonomic rank (such as genus). This can be easily tested by simulation (take some known complete genomes, randomly reduce their completeness and increase some contamination and

then do the taxonomic assignment). However, genus level results are reported in many places in the manuscript. Second, the MAGs are not comprehensive enough (the average mapping rate is only around 26%, Line 121). 16S rRNA gene sequencing should be conducted to analyze the community dynamics.

Other comments:

1. There many kinds of disturbances (such as temperate changes, shocks of salt/toxic substances, pH variations) in wastewater treatment plants. Therefore, I suggest that the "disturbance" in the title should be more specific.
2. Line 91-93: "To characterise the niche space of lipid-accumulating populations...we collected individual foaming sludge islets from..." Why collected foaming sludge islets instead of bulk sludge?
3. Figure 3a/3b: These two figures are quite misleading. They indicate that the presented genera accounted for 100% of the sludge bacterial community. The fact is that hundreds of other genera, which were not captured in this study, may exist in the sludge samples.

Point-by-point response

The original comments are in black font colour and our replies are in blue font colour. Moreover, we numbered the comments to improve readability, e.g., R.1.1. represents the first comment by the first Reviewer, R.2.1. represents the first comment by the second Reviewer. Line-numbers refer to the revised manuscript, i.e., the manuscript without track-changes. A pdf with detailed track-changes is also uploaded.

Reviewer comments:

Reviewer #1 (Remarks to the Author):

The manuscript by Harald et al. describes the use of a multi-omics approach to determine the phylogenetic and functional shifts that occur in a mixed-culture biotechnological process. Samples were collected from a municipal wastewater treatment plant over a period of 14 months, allowing detailed understanding of community fluctuations over time and responses to disturbances. Additional ex situ experiments were conducted with oleic acid feeding under different oxic conditions to evaluate the short-term responses to pulse disturbances. The study resulted in a large number of metagenome assembled genomes (MAGs) which were used to define different ecological niches based on the biochemical potentials of the MAGs.

R.1.1.

As an approach, this is an extremely elegant study that takes advantage of the current state-of-the-art in multi-omics technologies; approaches that can currently only be carried out by a few select laboratories, including that of the senior author.

We thank the Reviewer for the appreciation and recognition of our study, as well as the constructive comments. We share the Reviewer's assessment that our study reflects the current state of the art in multi-omic analyses of microbial communities. With our work, we describe a novel framework for multi-omic data integration and make this plethora of data available to the research community, which we believe will be extremely helpful to others to develop new methodologies and questions.

R.1.2.

Their work highlights the importance of interspecies niche complementarity for metabolism of complex substrates, such as encountered in wastewater plants. The main novel finding is that the MAGs could be divided into different fundamental niches. The resulting representative MAGs (rMAGs) were studied to determine their functional capacities based on their predicted gene functions. The temporal dynamics and activities of the rMAGS in the wastewater treatment plant over the 14-month period was determined based on read mapping of MGs and MTs, respectively. The study also distinguished between genes (KOs) that were expressed compared to those that were not expressed based on MT/MG ratios and

proteomics. This enabled them to study the plasticity of gene expression during shifts in environmental conditions.

The rest of the findings were more descriptive in nature and it is difficult to judge what is novel over what is already known about community structure and dynamics in wastewater treatment plants. The dominant *Microthrix* organism has been extensively studied in previous research.

We appreciate the Reviewer's comment. We have thus completely revised the Discussion in our revised manuscript and now further emphasize the novel aspects therein. In brief, to the best of our knowledge, our work is the first to reveal phenotypic plasticity and niche complementarity through the integration of multi-omics, and hence goes beyond the identification of fundamental niches. In particular, the clear delineation of the fundamental niches of constituent populations in contrast to the pronounced overlap in the realized niches demonstrates the importance to study the functions and activity of mixed microbial communities in greater detail than before. In addition to the specific novel results for the floating sludge system, the overall stability of the community and the reversion to a pre-disturbance state provide intriguing fields of research. We think the dataset itself is novel and highly relevant. We discuss our results for *Microthrix parvicella* due to its known importance in the context of lipid accumulation and being a keystone species in the wastewater treatment process. Our in situ study reveals that high abundance of individual genera, such as *Microthrix* among others, is not necessarily reflected in their mean expression levels. Moreover, we show that *Microthrix* overlaps with *Anaerolinea*, *Leptospira* and *Acinetobacter* populations in terms of their capacity to assimilate long chain fatty acids and available neutral lipids, thereby also providing novel insights beyond the dominant organism in this system. Our population-resolved multi-omic study allowed us to reveal differences in the individual adaptation strategies of known as well as uncharacterized organisms. We believe that many additional organisms of interest warrant detailed study and look forward to future studies of the herein presented data which is made openly accessible.

R.1.3.

What about the metaproteomics results? Lines 123-125 mentions how many spectra were obtained and that they were matched to MAGs. Where are the results? The only mention of the proteome data in the results is on line 300, but I couldn't find any description of the results or a figure. The only proteomics data were found on Supplementary Figure 12 for the ex situ experiment and that figure is not mentioned in the text.

We thank the Reviewer for highlighting this point. We now provide an extended description of the metaproteomic results in the revised manuscript (L127f, L285f, L299-305), and we included a new figure (Figure 5) which links metatranscriptomic and metaproteomic data as well as a new Supplementary Figure 5, which illustrates the taxonomic composition of the metaproteome and its change over time. A table of the time-resolved spectral counts for

genes is available in the archive of the processed dataset (<https://zenodo.org/record/3590397>) in the files:

- Databases/Proteomics/peptides_per_tp_regesfilt.tsv
- Output/GeneLevel_Expressiondata_combined_TimeSeries.RDS

Metaproteomics is the omics technology that proves the functional capacity of genes. Despite the fact that current metaproteomics are only able to cover the most highly expressed proteins, the data allows the identification of translated genes and therefore of their functional state. Importantly, we show in the revised manuscript (Figure 5) that several functional categories correlate in terms of metatranscriptomic and metaproteomic abundances. Furthermore, we used the metaproteomic data to define genes as active, even if absent in the metatranscriptomic data, which may be due to the lower half-life of transcripts.

Supplementary Figure 14 (Supplementary Figure 12 in the original manuscript) is referenced in the Supplementary Note 1 of the revised manuscript. Metaproteomic data is only available for the in situ dataset. We have now revised the caption of Supplementary Figure 14 to make this point clearer.

R.1.4.

Note that Supplementary figures 9,10, 11 and 12 are not mentioned in the manuscript.

In Supplementary Note 1, we highlight additional results for the expression analysis (MT and MP) of particular enzymes. Supplementary Figures 11, 12, 13, 14 (Supplementary Figures 9, 10, 11, 12 in the original manuscript) are referenced in Supplementary Note 1 only as they are central to it but not the main text.

Minor comments

R.1.5.

Assume that this is hypothesis generating research? I couldn't find a hypothesis that was tested.

We thank the Reviewer for this comment and clarify our hypothesis as well as main novel findings in the revised manuscript (most notably L78ff, L452ff). Our main hypothesis was that community resistance and resilience are a function of phenotypic plasticity and niche complementarity. With our novel framework for integrating time-series multi-omic data, we characterised fundamental and realised niche breadths of individual populations in situ, which we believe is key to understanding ecological processes within a microbial community, here, of lipid-accumulating organisms.

R.1.6.

By reducing the description of methods in the results section, it might be possible to add more results, for example from the proteomics data.

We appreciate the Reviewer's comment and realise that the Results section also contains descriptions of the methods. We reduced the respective text passages (L116-L119) in the Results to methodological aspects which we consider to be key for the reader to clearly follow and understand the novel framework presented herein. As described in our reply (R.1.3), we extended the description of the metaproteomic results in the revised manuscript. We also added results on the link between phylogenetic ancestry and functional enrichment, as suggested by Reviewer #3 (L157ff).

R.1.7.

Fig. 3 figure legend doesn't mention the metabolite ratio graphs.

We thank the Reviewer for highlighting this point. We have revised the figure legend accordingly.

R.1.8.

Figure 4 is labeled "abiotic factors" but is a heatmap of metabolites. Consider a better figure title and more complete figure legend.

We revised the figure legend and adjusted the figure title accordingly.

Reviewer #2 (Remarks to the Author):

This is a powerhouse manuscript that provides an unprecedented level of molecular and physiological detail into a wastewater treatment microbial community. The authors apply a wide range of -omics-based methodologies combined with state-of-the-art bioinformatics in a time-series manner to track dynamics in community composition and functionality. The manuscript is incredibly well written, the results are clear, and the main conclusions appear well supported by the data. All together, I commend the authors on their enormous efforts. While I am deeply impressed by the incredible amount of work and effort that went into this manuscript, I have several serious concerns.

We thank the Reviewer for their appreciation of our study design, our methodological approach, and the amount of work and effort which went into this.

1) DESCRIPTIVE/EXPLORATORY

R.2.1.

The study is very descriptive in nature. There is no clear question or hypothesis stated in the introduction, and the manuscript very much reads as an exploration of the meta-omics datasets.

I am left wondering why the authors performed the study in the first place, as the disturbance event was likely fortuitous. The lack of a clear question or hypothesis is not a problem for a

field-specific journal, such as ISME J, but I find it problematic for a broad audience journal where the reader may not be accustomed to such a highly descriptive study.

We recognize that these aspects could have been described better and, hence, have clarified our main hypothesis in the revised manuscript (L78ff.) as well as our main conclusions (L416, L430-433, L440ff, L453f). As described in our replies to Reviewer #1 (R.1.2., R.1.5.), our main hypothesis was that community resistance and resilience are a function of phenotypic plasticity and niche complementarity. The focus on lipid-accumulation and lipid-accumulating organisms is motivated by the profound implications in sustainable resource management, for which the underlying microbial ecology is understudied, especially compared to polyphosphate-accumulating organisms and glycogen-accumulating organisms.

We have been sampling this system as part of Luxembourg's national efforts on Sustainability Research. The disturbance event was not entirely unexpected as previous observations of the same system noted a characteristic shift in the floating sludge phenotype (<https://doi.org/10.1038/ncomms6603>) of lipid accumulators that has also been noted for other systems (<https://doi.org/10.2166/wst.2006.387>). Moreover, the fact that we were able to observe this shift in the first place is due to our extensive and continuous sampling scheme. How the system responds during and after this shift remained unstudied until now. We consider this an asset of our study and the associated data, with high relevance to the research community beyond the present work.

With respect to the comment related to "a broad audience journal", we would like to refer to a comment from the Editor who, in their decision letter, clarified that Nature Communications is not "a broad audience journal" in the same sense that Nature or Science are; broad appeal is not required for publication in Nature Communications, which is meant to publish 'papers representing important advances of significance to specialists within each field' (<https://www.nature.com/ncomms/about>). We believe that our work represents an important advance of significance to wastewater engineers specifically, as well as microbial ecologists in general. Our work is the first to integrate longitudinal metagenomic, metatranscriptomic, metaproteomic, and (meta-)metabolomic data for the study of wastewater-borne mixed microbial communities, and we are convinced that our newly developed methodology/framework is applicable to other ecosystems as well. We believe that the relevance and importance to the research community goes beyond what is described in the current work, which is why we openly provide the detailed meta-omics data for further study.

2) RATIONALE

R.2.2.

The authors use biotechnology as the rationale for their study (lines 26-27, 44-46, 411-412). However, it is unclear to me how any of the main outcomes could be used in practice. How would the knowledge gained help us to design, improve, and/or optimise a biological process? This needs explained.

The wastewater treatment process is one of the most common biotechnological applications. Detailed niche characterisation is crucial for process optimization and future reengineering of the process for the recovery of high-value biomolecules (<https://doi.org/10.3389/fmicb.2014.00047>). However, in the context of community resilience and resistance, our approach can be extended to other microbial communities, which may go beyond biotechnological applications. We thank the Reviewer for highlighting that this was not clear and have adapted the framing of our research question and rationale in the revised manuscript to reflect these broader implications (most notably L72ff, L87f, L456ff).

3) MECHANISTIC INSIGHT

R.2.3.

As is the typical problem with descriptive studies, I am missing mechanistic insights. For example, the authors observe a correlation between changes in certain substrates and community dynamics. What is the direction of causation? Do substrate changes cause community changes or vice versa? What is the underlying cause of the substrate changes? I am not convinced that the ex situ experiments help to understand this, as the in situ and ex situ substrates are different. As with point one above, the lack of convincing mechanistic insights is not a problem for a field-specific journal, such as ISME J, but is somewhat unsatisfying for a broad audience journal.

As highlighted by the Reviewer, this dataset provides an unprecedented level of molecular and physiological detail. The primary goal of this work was to better understand how phenotypic plasticity and niche complementarity are linked to resistance and resilience through integration of the four types of omics data. Therefore, the change serves as an example for a disturbance to the system and resolving the cause of the substrate change is beyond the scope of the present work. However, due to the observed oleic acid increase in situ, we designed the ex situ experiments to mimic the actual system by specifically including a pulse disturbance of excess oleic acid. Given that the ex situ results suggest distinct adaptation strategies upon disturbance, we believe that this supports our hypothesis that the substrate shift caused the community shift as described in the Discussion of the original manuscript. We have further emphasized this point in the revised manuscript (L443f). Concerning the comment on “broad audience journal”, we would like to refer to our previous reply on this aspect (R.2.1). Accordingly, we strongly believe that our work provides important advances of significance to specialists and is ripe to be mined by others, e.g., to perform mechanistic studies of specific pathways enabled by the longitudinal data and overall resistance and resilience of the system.

4) GENERALITY

R.2.4.

I assume that many of the observations are likely system specific. Which take-home messages are potentially generalisable and which are not? This needs discussed. If there

are no clearly generalisable outcomes, then I again question whether this manuscript is suitable for a broad audience journal.

We thank the Reviewer for this comment. We now further elaborate on the general applicability of our findings in the Discussion section of the revised manuscript (L456ff). In brief, we performed highly detailed characterisation of ecological niches using longitudinal meta-omics data to resolve adaptation strategies, and fundamental as well as realized niches in microbial populations. We believe that our novel framework is applicable to other ecosystems, including, but not limited to, host-associated microbiota, to better understand how similar disturbances and intermittent major regime shifts affect individual populations as well as the microbial community at large. We strongly believe that our work provides important advances of significance to specialists, e.g. in the wastewater treatment field, but also to microbial ecologists in general, and would like to refer to our previous replies on the Reviewer's "suitable for a broad audience journal" comments (R.2.1, R.2.3).

Reviewer #3 (Remarks to the Author):

R.3.1.

Herold and colleagues present a well-designed experiment to identify how fundamental and realized niches deviate from one-another for microbes in sewage sludge. By comparing fundamental niches based on metagenomics to realized niches based on repeated metatranscriptomics and metaproteomics, the authors concluded that the genomic repertoire was a poor predictor of relative abundance in response to changes in the abiotic environment. The authors then did a follow-up study in which they incubated sludge under a range of oxygen conditions to assess microbial plasticity on shorter time scales. This is an impressive dataset and a very interesting framing of the research at hand. I am unfamiliar with other papers which have taken such a holistic approach to understanding the interaction between ecology and biotechnology, and think this paper could be of broad interest to researchers in both these fields. I provide some comments below, primarily related to the role of phylogenetically-conserved core genes in occluding functional differences.

We thank the Reviewer for her appreciation of our study and the constructive comments. One of our goals with this work is to provide the research community with a framework of high resolution of multiple omes over time which we believe holds great potential to study interactions between microbial ecology and other factors, e.g., human health. We have addressed the comments regarding phylogenetically-conserved core genes in our replies below.

R.3.2.

MAGs:

Please report the quality data for each MAG used in the paper (contamination, completeness, taxonomy, coverage across the MAG etc as in doi:10.1038/ismej.2015.241). Table S2 just has the relative abundance of the MAGs.

We agree with the Reviewer that reporting the quality data for each MAG is important. We provided the respective information already in Supplementary Table S2 of our original manuscript. During the submission process, we uploaded an Excel file which was, however, converted into a PDF resulting in numerous pages and poor readability. We have contacted the technical support of Nature Communications and the files should now be available in the appropriate format (i.e. Excel) for review. We also added an additional Excel-sheet to Table S2, which is based on the 'Minimum Information about a Metagenome-Assembled Genome' (MIMAG) standards (<https://doi.org/10.1038/nbt.3893>) and details the quality criteria for all MAGs, starting with the 78 representative MAGs (rMAGs) characterised in detail within our study.

R.3.3.

Also, I would imagine that the presence and absence of genes (ie niche assignment) would be contingent upon genome completeness (and also perhaps niche assignment might actually correlate with completeness for biological reasons such as differences in relative abundance under different biotic conditions). Did you try re-running the analysis either with resampling each genome to the same completeness, or accounting for the differences in completeness/contamination in some other way? [L738]

We acknowledge the Reviewer's concerns regarding the completeness of the population-level genomes. Especially for clustering populations according to functional gene categories this is of course a highly relevant point. We would like to note, however, that the rMAGs studied in the context of niche assignment had a high average completeness (76.2%) and low degree of contamination (2.2%). We have not performed resampling experiments as we believe that, while being an interesting study in itself, niche assignments should be made based on as complete genomes as possible (<https://doi.org/10.1371/journal.pone.0000743>, <https://doi.org/10.1038/s42003-020-0856-x>). We assessed the effect of rMAG completeness on the example of FunC-4 which represents an "intermediary" cluster with rMAGs slightly less complete on average (73.2% vs 76.7% of FunC-1 to FunC-3), highlighting a marginal effect of completeness on FunC assignment. This effect, is however, expected to be amplified should less complete rMAGs be used, hence the definition of our stringent quality filtering criteria which were used in the analysis.

R.3.4.

MAG clusters:

L151: If genomes cluster by taxonomy, how do you attribute differences in niche related to functional genes rather than some longer, phylogenetically-conserved ability to show similar responses to a changing environment? I think it is essential to differentiate between functions which are enriched for because certain areas of the phylogenetic tree are enriched for, versus functions which are enriched for independent of the genetic background or whole-organism evolutionary history. The authors do discuss the taxonomic contribution in place (ex. L190, 203), but do not account for it in their conclusions.

One way to address this is to ask how the genomes of Actinobacteria which end up in cluster FunC1 differ from the complete pool of publicly-available Actinobacteria genomes (ie remove the “core” Actinobacteria genome)? Does genomic potential within this group constrain it to existing only in FunC1? Or do other members of Actinobacteria have many of the functional genes required to belong in other clusters, but the phylogenetic signal of the core genome is overwhelming? For instance, I think Actinobacteria genomes have been found to have nirK but be unable to complete denitrification (<https://doi.org/10.1371/journal.pone.0114118>). So this seems like the taxonomic (or phylogenetic) and functional niches cannot be separated, and that these clusters are only interesting insofar as they consist of organisms which have traits that cannot just be explained by ancestry. Another way to remedy this would be to run a phylogenetic PCA and see if you get the same “functional” clusters in the MAGs. Then you can think about how phylogeny defines fundamental niche, but does not constrain the realized niche. For instance, is the phylogenetically-corrected functional gene distance matrix correlated with the environmental response dissimilarity matrix for the taxa?

We very much welcome the Reviewer’s insight and agree that phylogeny and functional potential are linked. Motivated by the Reviewer’s comment, we performed additional PROCRASTES and PROTEST analysis to test how whole-genome based distance (computed using mash, <https://doi.org/10.1186/s13059-016-0997-x>) and functional niche assignment correlate. We now emphasize this point more strongly in the revised manuscript (L158ff). Based on our additional results, we found a strong and significant correlation (correlation 0.775, PROTEST p-value 0.001), which quantitatively supports our observations based on the taxonomic overlay in Figure 2. However, it also suggests that factors beyond phylogeny contributed to the formation of the fundamental niches. The observation that some branches of the tree are composed of rMAGs of different FunCs than their neighbors (Supplementary Figure 2b in the revised manuscript) further supports this and shows the importance of functional characterization. Importantly, using the functional omics data, we were interested in studying the realized niches and how these differ from the fundamental niches of the individual organisms. This is independent of what drives the fundamental niches originally. We followed the second suggestion of the Reviewer and also performed a phylogenetic PCA (using the phylogenetic tree inferred from the whole-genome based distances). The results of this additional experiment are shown below. While the individual

clusters are closer in the phylogenetic PCA plot than in the MDS plot in Figure 2, the individual FunCs remain apparent, again supporting that relevant factors beyond phylogeny are involved. We pursued this suggested experiment as it integrates the phylogeny into the two-dimensional embedding and performing an analysis on “core” and “non-core” genes would basically represent creating pan-genomes of individual taxa, such as the Actinobacteria (FunC-1), Bacteroidetes (FunC-2), Beta- and Gamma-Proteobacteria (FunC-3), and various taxa in FunC-4, which is beyond the scope of the current work.

Phylogenetic analysis of rMAGs and links to functional potential. (a) Phylogenetic PCA (phyl.pca function of the phytools package) of rMAG functional potential using (b) the phylogenetic tree inferred from the rMAGs' whole genome-based genomic distances. 95% ellipses are shown per FunC and percentages on the x-axis and y-axis show the amount of variance explained, respectively.

R.3.5.

Language:

I found the language was impenetrable in parts of the introduction. There are a lot of big words and complicated sentences, and it would be nice to provide the reader with some simpler language periodically so their mind can rest. The sentences between lines 70-75 were particularly challenging.

We thank the Reviewer for bringing this to our attention and have adjusted the language in the revised manuscript, especially in the Introduction, to further improve the manuscript's readability.

R.3.6.

Methods:

For people (like me!) who aren't familiar with how these anaerobic wastewater treatment chambers work, it might be useful to quickly state what the microbial community retention time is/how this thing works. So please move the text on L 393 regarding the rationale for sampling interval up to the beginning of the results. Does this flow/timeframe make the system more like a retentostat or a chemostat? Basically, do you need to account for the metabolites/proteins in the inflow in your data, or are the majority of compounds produced in-situ [L227-229 measures metabolites as substrates in the mat, not the incoming substrates, correct)?

We have followed the Reviewer's suggestion and moved L393. However, in accordance with R.1.6, we have moved this part to the Methods. The overall system per-se could be characterised as a retentostat. In this manuscript, we are investigating floating sludge islets which accumulate on top of the anoxic tank. As highlighted by the Reviewer, metabolite measurements were done on substrates in the islets which are likely a mixture from the inflow and produced in situ. Distinguishing between intra- and extra-cellular metabolites here adds to a broad characterisation of the available resource space at a given time-point. However, complementary approaches, e.g., using labeled substrates in situ, would be required to distinguish substrates produced within the microbial community from incoming substrates. While this is extremely interesting, this type of data was unfortunately not available for the in situ experiments and this work goes beyond the scope of the present study.

R.3.7.

The MAGs allow for population-level genomes, so are the observed patterns likely to be cell phenotypic plasticity, or due to shifts in microbial ecotypes within a population? Could this be tested with approaches like the one in Bendall et al. (doi:10.1038/ismej.2015.241)? Or are the genomes not of sufficient quality? Or are you mapping the metaT reads to a single representative MAG from across all the timepoints already?

We thank the Reviewer for the interesting suggestions. As outlined above (R.3.2.), we now provide additional information on the MAG quality and believe that the representative MAGs (rMAGs) have high degrees of completeness and low contamination degrees (please also see R.4.2., L120f). We indeed mapped the metaT reads against single rMAGs, i.e. the representative with the highest genome completeness and most consistent taxonomical classification of the population across all samples of the timeseries. We chose this approach to streamline the analysis. However, strain-level resolution of MAGs remains strongly limited by many factors, e.g., sequencing depth and/or quality of the assembly. Also, the metabolomic and metaproteomic data would be challenging to link to individual strains, albeit the latter could be partially resolved by identifying SNPs in proteins/peptides that correlate with SNPs in MAGs. However, this goes beyond the already extensive scope of the current work.

R.3.8.

L769 – how do the authors have a metagenome depth less than 1? Is this because they co-assembled metaT and metaG data to get the MAGs, so the gene was expressed without the gene being in the metaG data? If so, does this mean the MAGs cannot be trusted because of low coverage?

As the Reviewer correctly suggests, this can be a consequence of the co-assembly of metagenomic and metatranscriptomic data (<https://doi.org/10.1186/s13059-016-1116-8>). Moreover, the representative genomes are derived from individual timepoints, which means that depth-of-coverage values below 1 can occur in case the population that the rMAG represents is extremely lowly abundant at other timepoints. This is independent of the quality of the MAGs, i.e., the MAGs are of high quality as demonstrated by the high average completeness (76.2%) and low degree of contamination (2.2%) (L120f in the revised manuscript).

R.3.9.

I think your statistics about enriched genes need to be corrected for this lack of phylogenetic independence when comparing MAG and clusters of MAGs (L177-191).

As discussed above (R.3.4), we have performed extensive additional analyses to investigate the link between phylogeny and FunC assignment. These revealed that, while phylogeny is a strong factor, functional profiles clearly provide additional information. Enrichment analyses of gene functions in the clusters serve as a way to categorize the fundamental niches, implicitly including their respective phylogenetic background. This allowed us to identify two distinct niche types of particular importance for lipid-accumulation (L190ff), but also that several pathways were conserved across all FunCs (Figure 2c).

R.3.10.

Overall, this is a very nice paper. Good work!

We would like to thank the Reviewer again for the recognition of our work.

Reviewed by Grace Pold

Reviewer #4 (Remarks to the Author):

Herold et al. present an extensive omics analyses to reveal how microbial ecosystems respond to disturbance. The analyses include metagenomics, metatranscriptomics, metaproteomics and metabolomics based on both in situ sampling and ex situ lab experiments. Although a lot of data analysis work was conducted and presented in the

manuscript, I have strong concerns about the reliability of the analysis processes (especially metagenomic analyses) and the key findings of the study.

Main concerns:

R.4.1.

It's not very clear what are the key findings of this study. From the abstract section, I mainly got two main messages: (1) the authors uncovered four distinct fundamental niche types; (2) The change of community structure coincided with the changes of substrate availability. For the first one, it's not difficult to identify some different clusters/types from bioreactor bacterial community. After reading the manuscript, I'm not quite clear what is the significance of the four types identified in this study? How do they relate to the main functions (transformation or removal of organics, nitrogen and phosphorus) of the wastewater treatment bioreactors? As for the second message, I think it is a common sense for researchers and engineers in the field of wastewater treatment.

We thank the Reviewer for their observations and comments. We have now clarified in the manuscript that the aim of our work is to identify fundamental as well as realised niches in in situ measurements over time (R.1.2., R.2.1., R.2.4). To do so, our work is the first characterisation and integration of longitudinal, multi-meta-omic data of wastewater treatment borne microbial communities. This allowed us to test our hypothesis that community resistance and resilience are a function of phenotypic plasticity and niche complementarity, i.e., are based in organism-specific response strategies. Concerning the Reviewer's question on how the FunCs relate to the main functions in wastewater treatment plants, we highlighted that the different groups share potential for several transformations, e.g., nitrogen removal (L208f, Figure 2c), indicating that all FunCs seem to contribute to the main functions in wastewater treatment. Furthermore, we describe the main enriched functions of the individual FunCs in the revised manuscript (L185ff). Due to the relevance of lipids in the context of Sustainability Research, more specifically, energy recovery from wastewater, we focused on functions related to lipid metabolism (L190ff), e.g., TAG accumulation and PHA accumulation, but, of course, other metabolic processes are very relevant for future studies.

R.4.2.

2. It seems that the metagenomic analysis has serious flaws. (1) For the MAG construction, the authors applied a very coarse and arbitrary quality filtration criteria (>28% completeness, <20% contamination). To my knowledge, this is not acceptable at all. Such a low completeness and high contamination will dramatically affect the reliability of the downstream analysis. The minimum required information about a metagenome-assembled genome of bacteria and archaea can be found in this paper <https://www.nature.com/articles/nbt.3893> and many other published papers.

We thank the Reviewer for the critical comments on the quality of MAGs. We understand that the originally reported quality criteria for pre-filtering (<https://doi.org/10.1038/nmicrobiol.2016.180>) may have been misleading as this apparently provided an incomplete picture. We now report the rMAG quality by means of completeness (average of 76.2%) as well as contamination (average of 2.2%) in the Results of the revised manuscript (L120f). We believe that the values reported in the revised manuscript demonstrate the good quality of the reconstructed rMAGs. In the revised Supplementary Table 2, we now also report classifications according to MIMAG standards. Based on our detailed reasoning, we are of the opinion that the metagenomic analysis does not exhibit “serious flaws” but is founded on well-established definitions of MAGs and rMAGs.

R.4.3.

(2) Another concern about the metagenomic analysis is that the authors used many low-quality MAGs to investigate the bacterial community. To me, this is also not acceptable for the following two reasons. First, currently, there are no reliable methods to assign a MAG accurately to a low taxonomic rank (such as genus). This can be easily tested by simulation (take some known complete genomes, randomly reduce their completeness and increase some contamination and then do the taxonomic assignment). However, genus level results are reported in many places in the manuscript. Second, the MAGs are not reprehensible enough (the average mapping rate is only around 26%, Line 121). 16S rRNA gene sequencing should be conducted to analyze the community dynamics.

We thank the Reviewer again for their concern related to the quality of the MAGs in our study. As already mentioned in the preceding comment (R.4.2), we want to highlight the high completeness (average of 76.2%) as well as low contamination degrees (average of 2.2%) of the rMAGs in our study (L120f). We completely agree with the Reviewer that - in general - incomplete MAGs are a challenge in taxonomic classification and therefore a well-defined, broadly accepted definition of MAGs and rMAGs should be considered. The rMAGs in our study fulfil these criteria and, hence, we are confident that the taxonomic assignments are representative at their respective taxonomic ranks. Importantly, we used two distinct methods (conserved marker genes and whole-genome comparison) to support the assignments and also reported MAGs without confident assignment. Furthermore, we would like to emphasise that our analyses of functional complements and expression levels are not dependent on taxonomic classification in the first place, i.e. the FunCs are embedded and clustered in an unsupervised way. Taxonomic classification serves as means to put specific results into context (e.g. relative abundance or mean MT/MG ratios), but our study of fundamental and realized niches does not depend on taxonomic classification per se.

The aim of our work was to resolve the niche ecology of the recoverable populations on a functional level. It is well established that lowly or extremely lowly abundant populations are challenging to resolve, even with extreme sequencing depths. While we are aware that these organisms likely play roles within the community, we are confident that our analyses are

representative, as supported by previous 16S rRNA gene and metagenomic results of the same system (see below). Moreover, while the average metagenomic and metatranscriptomics mapping rates are 26 % and 27 % respectively, the mapping rates of identified peptides are much higher (43 % [L126]). We would also like to note that our metagenomic mapping rates are comparable to other current MAG-based studies, e.g. <https://doi.org/10.1038/s42003-020-0856-x> and <https://doi.org/10.1038/s41564-018-0176-9>.

We thank the Reviewer for the suggestion that 16S rRNA gene sequencing should be conducted to analyse the community dynamics. We would like to note, however, that the dynamics reported in our study are in line with previous reports on the study of the Schifflange WWTP (<https://doi.org/10.1038/ncomms6603>), which clearly demonstrated a high correlation between 16S rRNA gene sequencing data and metagenomic data albeit for a smaller set of timepoints. Moreover, the multi-omics data included herein (metagenomics, metatranscriptomics, metaproteomics, and (meta-)metabolomics) is more comprehensive than 16S rRNA gene sequencing data. Importantly, due to the increased resolution (time, sequencing depths, abiotic factors, number of omics), our present study enables insights into the system which were not possible in the previous study, most notably on resistance and resilience and how the system responds to disturbances. We make this dataset publicly available as we consider it to be an important resource for the research community and are confident that it will support the development of new microbiome analysis methods and the generation of new knowledge. As such, we believe that the analysis of fundamental and realized niches as well as the study of resistance and resilience presented in our current work provide an interesting entry point and will be complemented by future studies.

Other comments:

R.4.4.

1. There many kinds of disturbances (such as temperate changes, shocks of salt/toxic substances, pH variations) in wastewater treatment plants. Therefore, I suggest that the “disturbance” in the title should be more specific.

We agree with the Reviewer that there are many kinds of disturbances. Importantly, we discuss several disturbances, including seasonality, substrate change in the in situ experiment, and alternative oxygen concentrations in the ex situ experiment in the revised manuscript (L233ff, L286f, L433, L441-448). Hence, we believe that the current title clearly represents our results, which are not restricted to a single disturbance but also cover temporal fluctuations of abiotic factors, e.g. temperature and pH.

R.4.5.

2. Line 91-93: “To characterise the niche space of lipid-accumulating populations...we collected individual foaming sludge islets from...” Why collected foaming sludge islets instead of bulk sludge?

As shown in previous work in this system, foaming sludge islets are enriched in lipid-accumulating populations compared to bulk sludge (<https://doi.org/10.1016/j.copbio.2014.03.007>). While sometimes detrimental to WWTP operation, we believe that these populations are particularly interesting for strategies to recover high-value biomolecules (<https://doi.org/10.3389/fmicb.2014.00047>). However, a detailed understanding of niche ecology would be required for redesigning such processes. The sampling strategy was chosen accordingly.

R.4.6.

3. Figure 3a/3b: These two figures are quite misleading. They indicate that the presented genera accounted for 100% of the sludge bacterial community. The fact is that hundreds of other genera, which were not captured in this study, may exist in the sludge samples.

We agree with the Reviewer that potentially other genera, especially if present at very low abundances, may have been missed in our study. In Figure 3a/3b, we use this representation, as it is typical for the representation of compositional data. This plot enables the reader to see the overall stability in the resolved community composition, the composition shift, as well as the recovery. Moreover, we clearly reported the mapping rates in the Results section of the original manuscript. However, to ensure this information is also clearly linked to the figure, we revised the figure caption to highlight that the relative abundance of rMAGs grouped by genera is shown and included the mapping rates.

REVIEWERS' COMMENTS:

Reviewer #1 (Remarks to the Author):

I commend the authors on their careful revision of the manuscript. I do not have any additional major comments or concerns. I have a couple of minor comments below for the authors to consider:

line 89... I don't think that "essential" is appropriate in this sentence. In addition, the last sentence of introduction is too long and could be tightened up and clarified.

Line 122, 124, 145 and elsewhere: I am not sure that "rates" were measured; i.e. completion "rates" and "mapping rates". I recommend use of a different term.

Reviewer #2 (Remarks to the Author):

Reviewer 2

This is a great revision that largely addresses all of my concerns. I commend the authors again for an enormous effort and an exciting contribution to the field. I only have two comments that I would ask the authors to at least consider.

1) I greatly appreciate that the authors now explicitly state a main objective that goes beyond mere description: "We test the hypothesis that community resistance and resilience are a function of phenotypic plasticity and niche complementarity." Unfortunately, as is currently written, I'm not sure the authors actually tested this hypothesis in a rigorous manner. To do so, one would perhaps have to manipulate the amount of potential phenotypic plasticity and niche complementarity and then measure the consequences on resistance and resilience. I understand this is not possible with such a dataset and experimental design. I would therefore encourage the authors to rephrase this main objective (note, I do not think stating an explicit hypothesis is necessary. It could also be stated as a clear objective).

2) If the authors choose not to change the hypothesis, then I ask them to keep in mind that this hypothesis is not novel. I would advise them to cite relevant literature and provide an overview of our current understanding of the relationship between plasticity/complementarity and resistance/resilience.

Reviewer #4 (Remarks to the Author):

Overall the authors have addressed the comments they felt needed addressing. I have no further comments.

Point-by-point response

The original comments are in black font colour and our replies are in blue font colour. Moreover, we numbered the comments to improve readability, e.g., R.1.1. represents the first comment by the first Reviewer, R.2.1. represents the first comment by the second Reviewer.

Reviewer comments:

Reviewer #1:

R1.1. I commend the authors on their careful revision of the manuscript. I do not have any additional major comments or concerns. I have a couple of minor comments below for the authors to consider:

We thank the Reviewer again for the constructive feedback and appreciation of our work. We have integrated changes according to the Reviewer's minor comments in the newly revised manuscript.

R1.2. line 89... I don't think that "essential" is appropriate in this sentence. In addition, the last sentence of introduction is too long and could be tightened up and clarified.

We replaced "essential" with "important" and shortened the sentence to make it clearer.

R1.3. Line 122, 124, 145 and elsewhere: I am not sure that "rates" were measured; i.e. completion "rates" and "mapping rates". I recommend use of a different term.

We changed the terminology and now report "mapping percentages" and "completeness".

Reviewer 2:

R2.1. This is a great revision that largely addresses all of my concerns. I commend the authors again for an enormous effort and an exciting contribution to the field. I only have two comments that I would ask the authors to at least consider.

We thank the Reviewer for this positive feedback and have addressed the comments in the newly revised manuscript.

R2.2. 1) I greatly appreciate that the authors now explicitly state a main objective that goes beyond mere description: "We test the hypothesis that community resistance and resilience are a function of phenotypic plasticity and niche complementarity." Unfortunately, as is currently written, I'm not sure the authors actually tested this hypothesis in a rigorous manner. To do so, one would perhaps have to manipulate the amount of potential phenotypic plasticity and niche

complementarity and then measure the consequences on resistance and resilience. I understand this is not possible with such a dataset and experimental design. I would therefore encourage the authors to rephrase this main objective (note, I do not think stating an explicit hypothesis is necessary. It could also be stated as a clear objective).

We thank the Reviewer for highlighting this and acknowledge that the term “hypothesis” can convey different meanings depending on which field it is used in, e.g., the clear decision to reject/not reject the null hypothesis in statistical testing based on a p-value. We thus have removed this term from the newly revised manuscript and adjusted the text accordingly. Nevertheless, we believe that we have demonstrated that community resistance and resilience are a function of phenotypic plasticity and niche complementarity in our work.

R2.3. 2) If the authors choose not to change the hypothesis, then I ask them to keep in mind that this hypothesis is not novel. I would advise them to cite relevant literature and provide an overview of our current understanding of the relationship between plasticity/complementarity and resistance/resilience.

As stated above (R2.2), we have removed the term “hypothesis” in the newly revised manuscript.

Reviewer #4:

R4.1. Overall the authors have addressed the comments they felt needed addressing. I have no further comments

We thank the Reviewer again for the constructive feedback, which helped improve our manuscript.